# Systems approaches identify the consequences of monosomy in somatic human cells

Narendra Kumar Chunduri [1], Paul Menges[1], Xiaoxiao Zhang[2], Angela Wieland[1], Vincent Leon Gotsmann[3], Balca R. Mardin [4], Christopher Buccitelli[4], Jan O. Korbel [4], Felix Willmund [3], Maik Kschischo [2], Markus Raeschle[1] & Zuzana Storchova [1✉]

Chromosome loss that results in monosomy is detrimental to viability, yet it is frequently observed in cancers. How cancers survive with monosomy is unknown. Using p53-deficient monosomic cell lines, we find that chromosome loss impairs proliferation and genomic stability. Transcriptome and proteome analysis demonstrates reduced expression of genes encoded on the monosomes, which is partially compensated in some cases. Monosomy also induces global changes in gene expression. Pathway enrichment analysis reveals that genes involved in ribosome biogenesis and translation are downregulated in all monosomic cells analyzed. Consistently, monosomies display defects in protein synthesis and ribosome assembly. We further show that monosomies are incompatible with p53 expression, likely due to defects in ribosome biogenesis. Accordingly, impaired ribosome biogenesis and p53 inactivation are associated with monosomy in cancer. Our systematic study of monosomy in human cells explains why monosomy is so detrimental and reveals the importance of p53 for monosomy occurrence in cancer.

[1] Department of Molecular Genetics, TU Kaiserslautern, Kaiserslautern, Germany. [2] University of Applied Sciences Koblenz, Remagen, Germany. [3] Group Genetics of Eukaryotes, TU Kaiserslautern, Kaiserslautern, Germany. [4] Genome Biology Unit, European Molecular Biology Laboratory (EMBL), Heidelberg, Germany. ✉email: storchova@bio.uni-kl.de

Human cells contain two sets of homologous chromosomes and maintaining this diploid chromosomal content is essential for their survival and proliferation. Errors in chromosome segregation that lead to a gain or a loss of chromosome, so called whole chromosomal aneuploidy, are poorly tolerated in humans[1]. Most cells arrest or die soon after chromosome missegregation and the aneuploid cells that escape this fate do not proliferate as efficiently as the diploid counterparts[2–5]. The low fitness of aneuploid cells is well documented by the fact that non-diploid karyotype is among the main causes of spontaneous abortions[1]. Most of our knowledge on the consequences of aneuploidy has been obtained by analysis of cells with extra chromosomes. Gain of even a single chromosome leads to marked physiological changes independent of the chromosome identity, and are conserved in various species from yeasts to human cell lines (reviewed in refs. [6,7]).

Much less is known about the consequences of chromosome losses, or monosomy, mainly due to the lack of a defined model system and its detrimental effect on proliferation and viability. Murine monosomic embryos die significantly earlier than trisomic ones[8] and studies from in-vitro fertilized human monosomic embryos indicate that monosomy drastically impairs their viability and implantation potential[9,10]. Only rare partial monosomies or micro-deletions are viable, but with severe pathological consequences[11]. These detrimental phenotypes are likely due to haploinsufficiency of some genes as well as due to unmasking of recessive mutations[12–14]. Monosomy of X chromosome is also lethal and the small percentage of embryos that survive suffer from severe pathological consequences (so called Turner syndrome)[15].

On the other hand, loss of an entire chromosome or its arm occurs in a substantial fraction of cancer types and associates strongly with hematopoietic cancers[16]. Recurrent whole chromosome or arm level deletions are frequent in specific tumors, such as 1p deletion in neuroblastoma, 3p deletion in lung cancer, or loss of 7, or 7q, in myeloid malignancies[16,17], suggesting an important role of chromosome losses in cancer pathogenesis. It has been proposed that chromosome loss may instigate cancer initiation due to haploinsufficiency of tumor suppressor genes. For example, frequent chromosome 17p deletion in broad spectra of tumors is attributed to loss of *TP53* tumor suppressor gene. However, modeling 17p loss in mice identified several other tumor suppressor genes that cooperate to generate more aggressive tumors[18].

In this work, using monosomic cells derived from the human immortalized hTERT-RPE1 lacking *TP53*, we analyze the impact of monosomy on proliferation, genomic stability, and how chromosome loss shapes the global transcriptome and proteome using multiomics approaches. We show a consistent reduction of cytoplasmic ribosomal proteins and impaired protein translation in all monosomic cell lines. Reintroduction of *TP53* to monosomies further impairs their proliferation due to p53 pathway activation. Analysis of The Cancer Genome Atlas (TCGA) and Cancer Cell Lines Encyclopedia (CCLE) databases reveals a strong association of monosomy with p53 inactivation and ribosomal pathway impairment. Our systematic analysis provides insight into the consequences of chromosome loss in somatic human cells.

## Results

**Monosomy impairs cell proliferation and leads to genomic instability.** To study the consequences of monosomy in human cells, we analyzed monosomic cell lines derived from RPE1, a human hTERT-immortalized, retinal pigment epithelial cell line. The used monosomic cell lines arose from a chromosome

missegregation in cells lacking p53 due to *TP53* deletion by CRISPR-Cas9 or by TALEN (see "Methods" section for details), or due to expression of shRNA against *TP53*[2] (Fig. 1a). The chromosome losses were identified by whole-genome sequencing of single-cell derived clones (Fig. 1b[2]) and chromosomal painting further validated the karyotypes (Fig. 1c and Supplementary Table 1). The monosomic cell lines were named **R**PE1-derived **M**onosomy (RM), followed by the number of the monosomic chromosome, i.e., RM 13 for monosomy 13; cell lines with the shRNA mediated knock-down of p53 are additionally labeled with KD (i.e., RM KD 13, Fig. 1a and Supplementary Table 1). Some monosomic cell lines showed variable partial or mosaic chromosome gains or losses, e.g., a gain of 22q in RM 13 (Fig. 1b). These small chromosomal changes occurred likely due to the increased genomic instability of monosomic cells (see below) and the relaxed checkpoint control owing to the p53 loss[2,3].

All monosomic cells proliferated slower than their respective parental diploids and this defect correlated with the number of open reading frames present on respective monosome (Fig. 1d and Supplementary Fig. 1a). While parental RPE1 cells do not form colonies on soft agar, deletion of *TP53* increased the cellular capacity of anchorage independent growth in RPE1 cell line. The monosomic p53 deficient cells formed significantly fewer colonies than the diploid parents (Fig. 1e). Notably, the loss of chromosome X from RPE1 impaired proliferation similarly as the loss of an autosome. In XX cells, one copy of the chromosome X is transcriptionally inactivated by XIST-mediated silencing, but there are approximately 100–130 genes located on chromosome X known to escape the X-inactivation[19]. Our observation therefore suggests that the loss of these few escapees is sufficient to impair cellular proliferation.

Cell cycle profiling by flow cytometry showed no uniform changes in cell cycle distribution in monosomic cell lines (Supplementary Fig. 1b). Strikingly, five out of seven monosomic cell lines suffered from significantly more mitotic errors than the parental cells, as manifested by the occurrence of lagging chromosomes and micronuclei (Fig. 2a, b). Approximately 20% of the micronuclei stained positively for centromere (CENP+), suggesting that only a small fraction contained whole missegregated chromosome rather the micronuclei enclosed a chromosome fragment. The fraction of CENP+ micronuclei was similar in diploids and monosomic cell lines (Supplementary Fig. 1c). Accordingly, the time from nuclear envelope break down till anaphase increased on average from 28 in diploids to 38 min in monosomic cells with exception of RM X (Fig. 2c). This finding also suggests that the spindle assembly checkpoint remains intact in monosomies. Significantly elevated occurrence of chromatin bridges, which form during anaphase due to under-replicated or incorrectly repaired DNA, was observed in three out of four monosomic cell lines lacking *TP53*; RM X and the p53 KD showed no significant changes (Fig. 2d and Supplementary Fig. 1d). Three out of four analyzed monosomic cell lines lacking *TP53* showed an accumulation of γH2AX foci that mark the double strand breaks (Fig. 2e, f). However, the DNA damage response was not activated, and the replication proteins were expressed at normal levels, as documented by immunoblotting of Chk1, pChk1 S345, RPA32, pRPA32 S4/S8, MCM2, and MCM7 (Supplementary Fig. 1e–h). Importantly, genomic and chromosomal stability was not impaired by the loss of chromosome X.

Chromosome gains cause increased proteotoxic stress in eukaryotic cells, manifested by impaired protein folding and increased sensitivity to inhibitors of protein folding and autophagy (17-AAG and chloroquine, respectively), as well as by increased autophagic and proteasomal activity[20–23]. In contrast, the sensitivity to 17-AAG was not significantly altered in monosomic cells and the expression of heat shock and

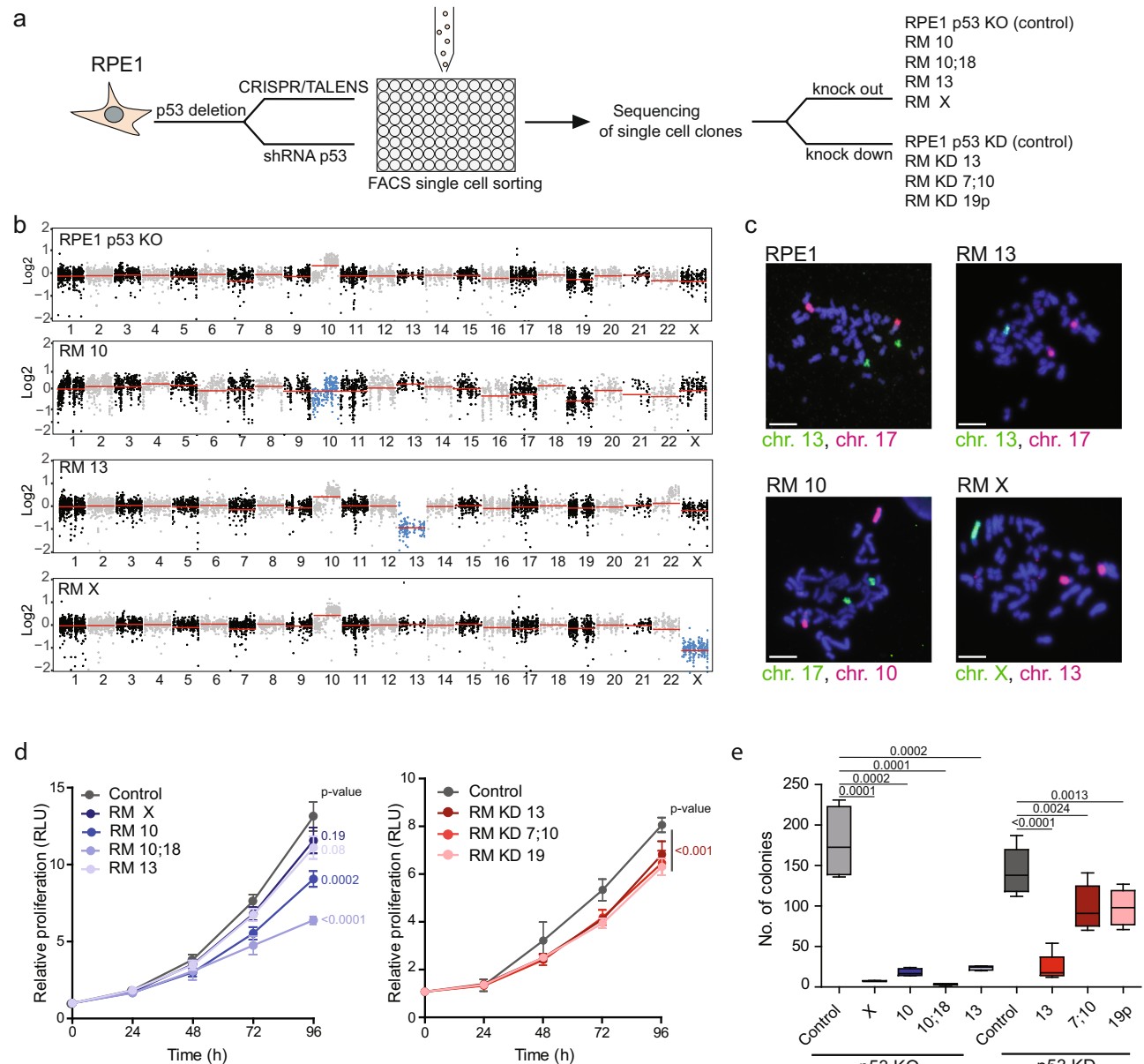

**Fig. 1 Chromosome loss in p53 deficient cells reduces proliferation and anchorage independent growth on soft agar. a** Schematic depiction of the construction of monosomic cells. p53 was mutated or depleted in RPE1-hTERT cell line via CRISPR/Cas9, TALENS, or shRNA expression. Clones arising from single cells were subjected to whole-genome sequencing and monosomic clones were selected to be used in further experiments. **b** Read depth plots of all chromosomes in control and RM samples. Chromosome losses are marked in blue. Red lines indicate the copy number of each chromosome. Note that the parental RPE1 contains an extra copy of 10q that is preserved in all monosomic derivatives. **c** Chromosomal paints of monosomic cell lines. The painted chromosomes are indicated with respective colors. Representative images of one experiment are shown. Scale bar—10 μm. **d** Proliferation curves of monosomic cell lines in comparison to respective diploid controls. All the values are normalized to day 0. Each point represents the mean ± SEM. Number of biological replicates: RPE1 P53 KO control = 7, RM X = 7, RM 10 = 7, RM 10;18 = 7, RM 13 = 7; RPE1 P53 KD control = 5, RM KD 13 = 5, RM KD 7;10 = 5, RM KD 19p = 5). RLU-relative luminescence unit. Statistical significance is calculated using linear regression analysis; p-values are shown in the plots. **e** Quantification of number of colonies growing on soft agar. The median is shown as a black bar and Whiskers are chosen to show the 1.5 of the interquartile range. Number of biological replicates: RPE1 P53 KO control = 4, RM X = 4, RM 10 = 4, RM 10;18 = 4, RM 13 = 4; RPE1 P53 KD control=8, RM KD 13 = 8, RM KD 7;10 = 8, RM KD 19p = 8. Statistical analysis was performed using one-tailed unpaired T-test; p-values are shown on the plot. Source data are provided in a Source Data file.

autophagy proteins was not uniformly deregulated (Supplementary Fig. 2a–e). Taken together, a loss of single chromosome leads to proliferation defects and to heterogeneous changes in maintenance of genomic instability, but does not trigger replication defects and proteotoxic stress, implying that the affected molecular processes do not parallel the previously identified effects of chromosome gains.

**Gene expression changes caused by monosomy are partly buffered at both mRNA and protein levels.** To obtain a global view of the expression changes induced by chromosome loss, we performed transcriptome and proteome analyses of monosomic cell lines RM 10;18, RM 13, RM 19p, and RM X. The obtained transcripts were normalized to the parental control cell line and visualized as the $\log_2$ fold changes (FC) according to the

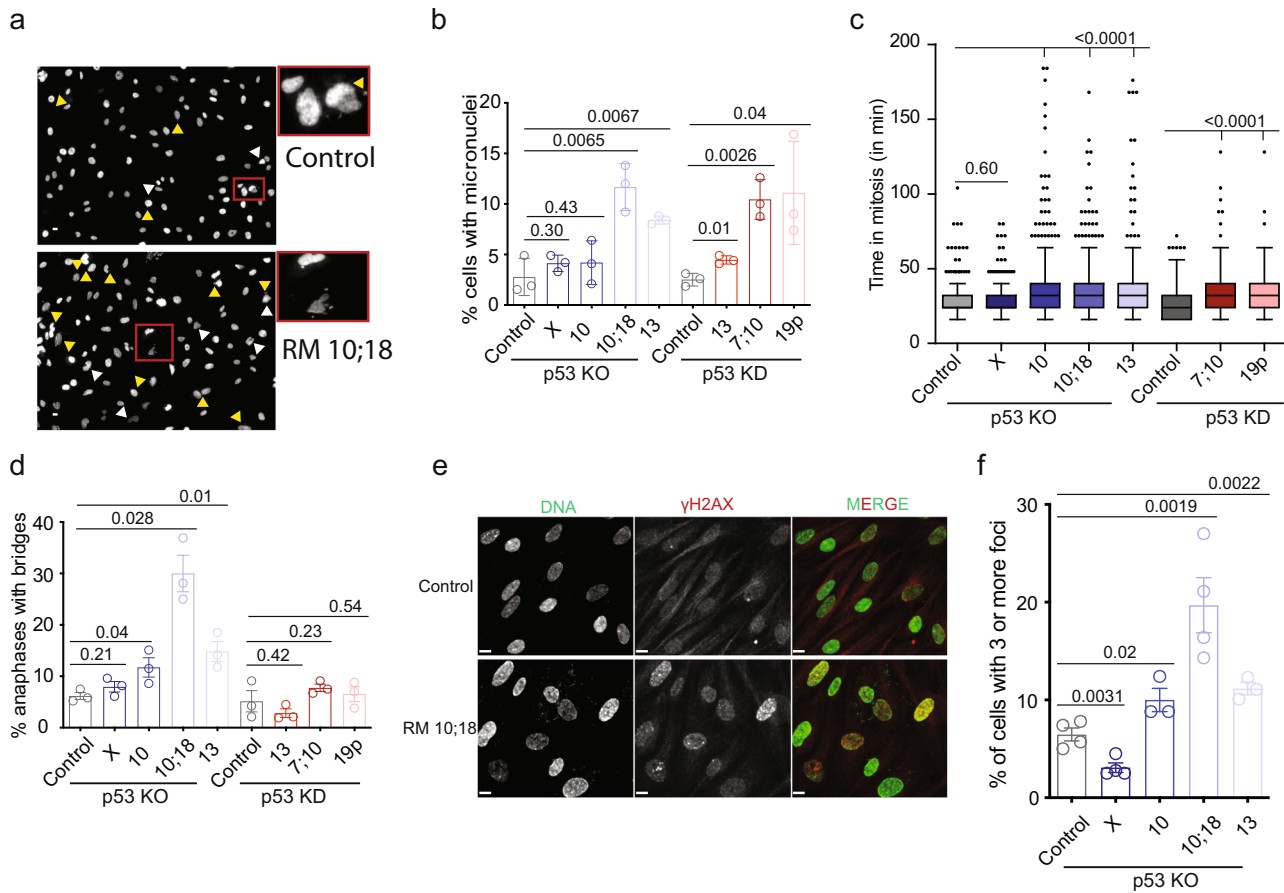

**Fig. 2 Genomic instability is increased in monosomic cell lines. a** Representative images of cells with micronuclei and nuclei structure abnormalities. Inset shows the enlarged field (white arrowhead—nuclear abnormalities, yellow arrowhead—micronuclei). Three independent experiments were performed. Scale bar—10 μm. **b** Percentage of cells with micronuclei. Bar graphs display the mean ± SEM of three independent experiments. Number of cells analyzed (RPE1 P53 KO control = 1879, RM X = 2657, RM 10 = 3928, RM 10;18 = 2196, RM 13 = 2065; RPE1 P53 KD control = 2082, RM KD 13 = 2155, RM KD 7;10 = 1922, RM KD 19p = 2149). Statistical analysis was performed by comparing to the respective controls using two-tailed unpaired T-test; p-values are shown on the plot. **c** Quantification of time spent in mitosis. Time from nuclear envelope breakdown (NEBD) until the end of anaphase was measured. The median is shown as a black bar and whiskers are chosen to show the 1.5 of the interquartile range for four independent experiments. Number of cells analyzed (RPE1 P53 KO control = 358, RM X = 1161, RM 10 = 729, RM 10;18 = 446, RM 13 = 572; RPE1 p53 KD control = 754, RM KD 7;10 = 242, RM KD 19p = 149). Statistical analysis was performed by comparing to the respective controls using two-tailed unpaired T-test; p-value for all the comparisons was <0.0001 as shown on the plot. **d**. Percentage of anaphases with bridges. Bar graphs display the mean ± SEM of three independent experiments. Number of anaphases analyzed (RPE1 P53 KO control = 201, RM X = 198, RM 10 = 209, RM 10;18 = 162, RM 13 = 207, RPE1 P53 KD = 197, RM KD 13 = 144, RM KD 7;10 = 182, RM KD 19p = 189). Statistical analysis was performed by comparing to the respective controls using two-tailed unpaired T-test; p-values are shown on the plot. **e** Representative images shows immunofluorescence staining of the DNA damage marker γH2AX in control and RM 10;18 cell lines. DNA is stained with sytox green (green) and γH2AX (red). The figures are representative of three independent experiments. Scale bar—10 μm. **f** Quantification of percentage of cells with ≥3 γH2AX foci per cell. Bar graphs display the mean ± SEM of four independent experiments, except for RM 10 and RM 13 with three independent experiments. Number of cells analyzed (Control = 2148, RM X = 2217, RM 10 = 1619, RM 10;18 = 1372, RM 13 = 1029). Statistical analysis was performed by comparing to the respective controls using one-tailed unpaired T-test; p-values are shown on the plot. Source data are provided in a Source Data file.

chromosome positions (Fig. 3a and Supplementary Fig. 3a, c, e). In the next step, we analyzed the proteome changes in monosomic cells. To this end, we used Tandem Mass Tag (TMT) labeling-based multiplexed quantitative proteomics to quantify the $\log_2$ FC compared to the parental cell line. The obtained values were normalized to the parental control cell lines and visualized as the $\log_2$ fold changes, similarly as the transcriptome data (Fig. 3b and Supplementary Fig. 3b, d, f and Supplementary Data 1). The TMT labeling strategy suffers from ratio compression[24], which means that the quantified fold changes are often smaller than the real values. Therefore, we also performed a label free protein quantification (LFQ). Comparison of LFQ data with TMT data revealed a high correlation between the values obtained with these different approaches (Supplementary Fig. 4a),

but a lower coverage of identified proteins was achieved with LFQ. Therefore, we used the TMT-determined values for the subsequent analysis.

The proteome analysis revealed, similarly as the transcriptome analysis, a reduced abundance of proteins encoded on the monosomes (Fig. 3a, b and Supplementary Fig. 3a–f). To elucidate whether the gene expression scales linearly with the gene copy number, we analyzed all genes for which both transcript and protein values were quantified (Supplementary Data 1). We compared the distribution of mRNA and protein $\log_2$FC for genes encoded on the monosomic and disomic chromosomes in individual cell lines. If all genes on monosomic chromosome were expressed according to their DNA copy number at 50%, then the median of $\log_2$FC should be −1.

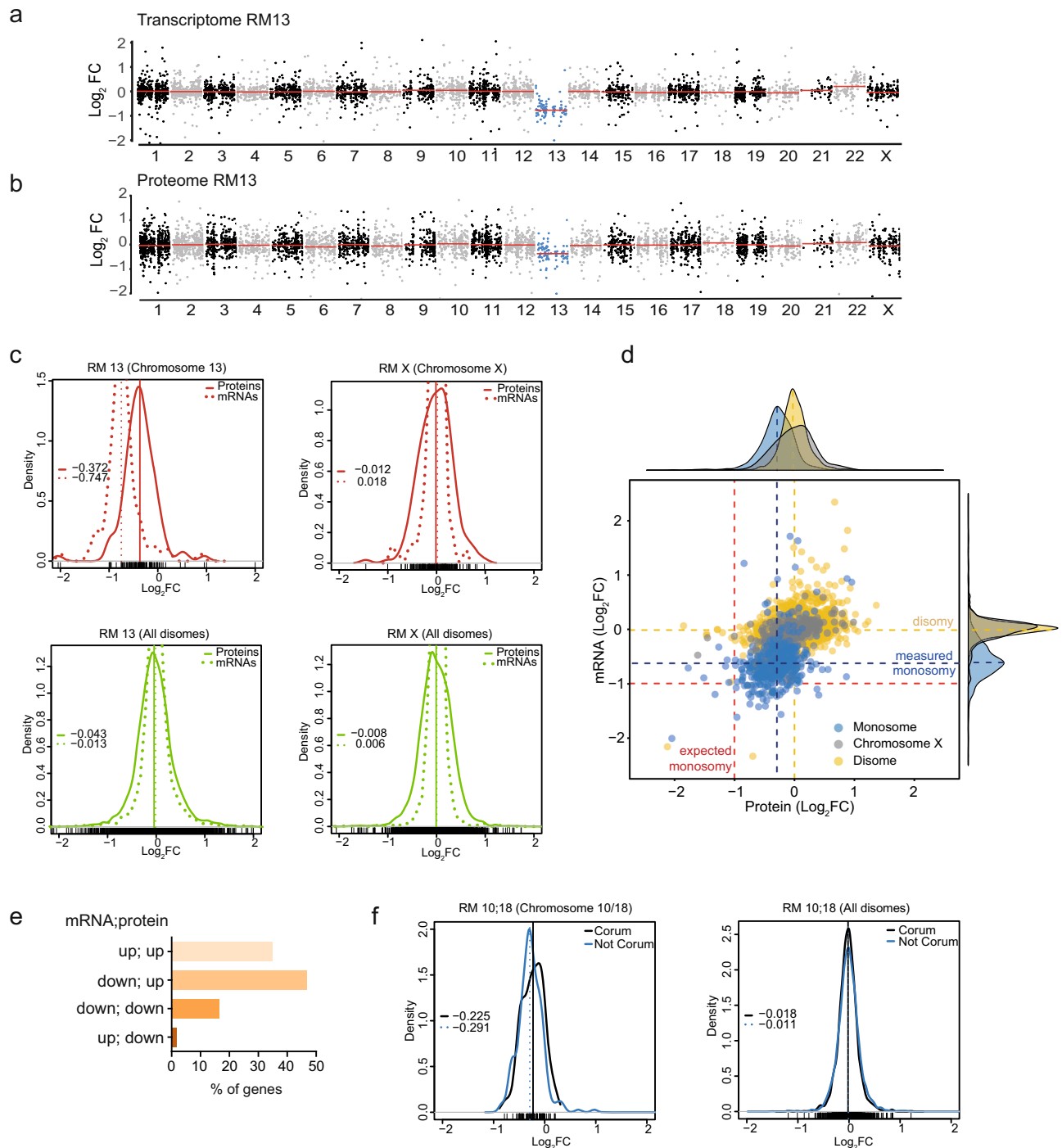

**Fig. 3 Expression of genes encoded on the monosomes is adjusted by transcriptional and posttranscriptional mechanisms. a, b** The relative abundance of mRNAs and proteins of RM 13 normalized to diploid isogenic parental control were plotted according to their chromosome location. The monosomic chromosome is marked in blue. Red line depicts the median for each chromosome. **c** Overlays of mRNA (dashed line) and protein (solid line) density histograms. Values of respective medians are plotted in the graph. Upper panels (in red) represent the monosomic chromosome; lower panel (in green) display all other, disomic chromosomes. **d** Scatter plot showing the log$_2$ fold change (FC) of mRNAs and proteins encoded on monosomes (blue), disomes (yellow) and chromosome X (gray). The marginal density histograms show the distribution of respective mRNAs and proteins. The expected median fold change of monosomic genes is marked by red dashed lines. The measured median fold changes of monosomic and disomic genes is marked by blue and yellow dashed lines, respectively. **e** Bar plot shows the percentage of genes assigned to categories according to log$_2$FC with a cut-off set to −0.5. up; up (both mRNA and protein more than −0.5), down; up (RNA < −0.5, protein > −0.5), down; down (both mRNA and protein log$_2$FC less than −0.5), up; down (mRNA > −0.5, protein < −0.5). **f** Density plots of subunit of macromolecular complexes as defined by CORUM database compared to non-CORUM proteins. Left panel displays the monosomic chromosome; the right panel shows the distribution for diploid chromosomes for RM 10;18. The median values are plotted in the graph.

However, analysis of RNA-seq data showed that the abundance of mRNA encoded on the monosomes (chromosomes 13, 10;18 and 19p, respectively) did not decrease to the expected levels. Instead, the mRNA median of monosomic gene expression ranged from −0.5 to −0.75 (Fig. 3c, upper panels, dotted line, S4b, Supplementary Data 2). The median of mRNA expression from X chromosome in RM X remains close to zero, as expected (Fig. 3c). The relative abundance of proteins encoded on monosomes was further increased (Fig. 3c, upper panels, full line, S4b, Supplementary Data 2). The distribution of mRNA and protein abundances for genes encoded on disomes was comparable in all monosomic cell lines (Fig. 3c, lower panels, Supplementary Fig. 4b, and Supplementary Data 2). When we combined the expression values from all genes encoded on the monosomic chromosomes, excluding chr. X, the median of $\log_2$FC of protein abundance was −0.25, while the median of all corresponding transcripts was −0.59, both values being significantly higher than the expected −1 (Fig. 3d). We then assigned the monosomically encoded genes into four categories with a cut-off set at −0.5: down; down (both mRNA and protein $\log_2$FC less than −0.5), up; down (mRNA > −0.5, protein < −0.5), down; up (RNA < −0.5, protein > −0.5) and up; up (both mRNA and protein more than −0.5). This showed that the expression of approximately 30% of the genes encoded on the monosomes were adjusted to diploid levels transcriptionally (up; up); additional 45% were adjusted posttranscriptionally (down; up) (Fig. 3e). Consequently, less than 20% of monosomically encoded genes were expressed at a relative abundance lower than −0.5 $\log_2$FC of the parental control. Of note, analysis of LFQ dataset provided comparable results (Supplementary Fig. 4c, d). Together, our data suggest the existence of mechanisms that alleviate the effects of monosomy on gene expression at both transcriptional and posttranscriptional levels.

Previous work on trisomic human cells uncovered that the protein abundance of approximately 25% of proteins encoded on the trisomes was adjusted to closely match the disomic abundance, and this was most strikingly prominent for proteins subunits of macromolecular complexes[25,26]. Comparison of the abundance of subunits of macromolecular complexes, as defined in the CORUM database, with the non-CORUM proteins revealed only a subtle shift towards diploid levels, suggesting that this mechanism is not important in monosomic cells (Fig. 3f and Supplementary Data 2). There was no protein dosage adjustment of membrane proteins, cytosolic proteins or any other biological functions, cellular compartments or further categories as defined by Panther, GO, or Perseus databases. Taken together, the effects of reduced gene copy numbers in monosomic cells are mitigated by both transcriptional and posttranscriptional mechanisms in human cells, but the mechanism remains to be investigated.

**Genome-wide changes of gene expression in response to chromosome loss.** We next analyzed the global gene expression changes in monosomic cells. Identification of proteins that were uniformly deregulated in response to monosomy revealed only five upregulated and 13 downregulated proteins with $\log_2$FC greater than 1.5 or less than −1.5 folds and shared in at least two different monosomic cell lines (Fig. 4a, b and Supplementary Data 1). Although the correlation between mRNA and protein abundance was rather modest (Supplementary Fig. 5a), 2D annotation enrichment algorithm that identifies deregulated pathways[27] showed a similarity between transcriptome and proteome of individual monosomic cell lines (Fig. 4c and Supplementary Fig. 5b, c, and Supplementary Data 3). Further comparison of proteome revealed that the individual monosomic

cell lines deregulated mostly a unique set of pathways (Fig. 4d and Supplementary Fig. 5d, e). For example, "MHC I and II protein complex" and "Interferon gamma mediated signaling" were upregulated in RM 10;18, but downregulated in RM 13 (Fig. 4d and Supplementary Data 3), which we confirmed by qPCR analysis (Supplementary Fig. 6a, b). Similarly, "O-glycan processing" was upregulated in RM X, while down regulated in RM 10;18 (Supplementary Fig. 5d and Supplementary Data 3). Strikingly, one of the few uniform and consistent deregulation identified in all monosomic cell lines was the downregulation of cytosolic large and small ribosomal subunit and translation (Fig. 4d, e and Supplementary Fig. 5d, e and Supplementary Data 3). Therefore, we continued with a detailed analysis of ribosomal metabolism and translation changes in monosomic cells.

**Ribosome biogenesis and translation is impaired in monosomic cells.** To understand the impact of reduced translation and ribosome gene expression, we determined the translational activity by puromycin-incorporation assay in monosomies. In this assay, proliferating cells were treated with a short pulse of the antibiotic puromycin (15 min, 10 μM) that is incorporated into nascent polypeptide chain. Immunoblotting of cell lysates with an anti-puromycin antibody therefore approximates the translation rate in the cells. Indeed, translation was significantly decreased in all tested monosomic cell lines (Fig. 5a, b). Eukaryotic protein synthesis is regulated via the mTOR pathway[28]. Immunoblotting of the mTOR target p70S6K and its phosphorylation revealed no reduction of mTOR activity in monosomies (Fig. 5c, d). Recently, reduced protein synthesis in Down syndrome mice models and Down syndrome patients' derived cell lines was attributed to integrated stress response (ISR)[29]. Increased phosphorylation of eukaryotic initiation factor 2 alpha (eIF2α) is a marker for ISR. However, immunoblotting of p-eIF2α showed no uniformly increased eIF2α phosphorylation in monosomies (Fig. 5c, d). Thus, the decreased translational activity in monosomic cells does not occur due to reduced mTOR activity or activation of ISR.

To elucidate the nature of the translation defect in monosomic cells, we performed polysome profiling to determine the fraction of polysomes, monosomes (composed of one ribosome residing on an mRNA), as well as unassembled small (40S in eukaryotes, SSU) and large (60S in eukaryotes, LSU) ribosomal subunits. Strikingly, monosomic cell lines accumulated more individual ribosomal subunits than the parental diploid cell line and the ratio of unassembled large and small ribosomal subunits was altered (Fig. 5e, f). We observed a reduction of SSU and accumulation of LSU in RM 10 when compared to diploid control, which is consistent with RPS24 (small ribosomal subunit protein 24) being the only RPG encoded on Chr.10 (Fig. 5e, f). In RM 13, the monosomic chromosome 13 encodes only one RPG RPL21 (a subunit of LSU) and, accordingly, we observed increased levels of free SSU. Interestingly, the monosomic cell lines appeared to contain more heavy polysomes than the parental control. While the increased polysome peak suggests increased ribosomes on mRNA, whether this is due to compensatory increase in the translation or caused by ribosome stalling leading to accumulation of multiple ribosomes on mRNA remains to be evaluated in future.

Each chromosome, with exception of chromosomes 7 and 21, carries one or more genes encoding ribosomal proteins (RPG—ribosome protein genes) that are known for their haploinsufficiency[30]. Therefore, we hypothesized that the translation in monosomic cells is impaired due to reduced ribosomal biogenesis caused by RPG haploinsufficiency. To test this hypothesis, we analyzed the consequences of RPL21 depletion,

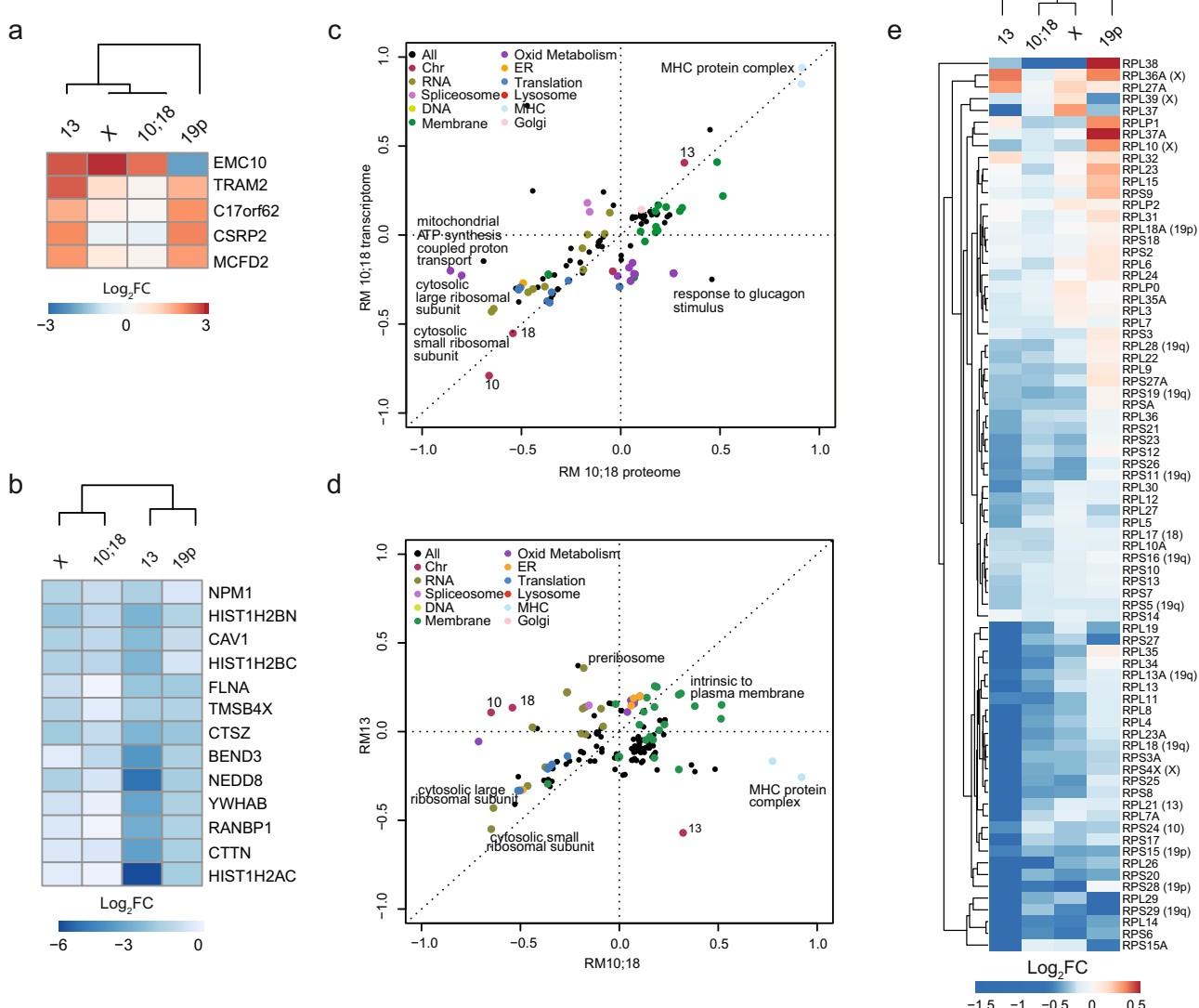

**Fig. 4 Transcriptome and proteome comparisons reveal pathway changes in monosomies. a, b** Heat map of upregulated and downregulated, respectively, proteins that were commonly altered in at least two monosomic cell lines. Log$_2$FC of monosomy compared to diploid are depicted. **c** Two-dimensional annotation (2D) enrichment analysis based on the protein and mRNA changes in the monosomic cell line RM 13 relative to the diploid parental cell line. Each dot represents one category (GOBP, GOCC and chromosome location, see Supplementary Data 1). Colors mark groups of related pathways as described in the inset. Axis-position represents scores of the pathways; negative values indicate downregulation; positive values indicate upregulation. Benjamini–Hochberg FDR Threshold <0.02. **d** 2D enrichment analysis comparing proteome of RM 13 with RM 10;18. **e** Deregulation of the expression of genes coding for ribosomal proteins. In brackets is the chromosome number for relevant RPGs.

which is the only RPG encoded on chromosome 13. Using siRNA, we titrated the levels of RPL21 in parental diploid RPE1 to approximately 50% of the wild type abundance, which resulted in reduced puromycin incorporation, similarly as observed in RM 13 (Fig. 5g–j). No significant changes to phosphorylation of p70S6K (mTOR target) and eIF2α (ISR marker) was observed upon RPL21 depletion (Fig. 5g). Thus, in agreement with the observation in monosomic cells, the general translation machinery is not inhibited. Knocking down *RPS24*, the only RPG encoded on chromosome 10, reduced the abundance of this protein to the levels observed in monosomy 10 cell lines (RM 10 and RM 10;18) and resulted in similar translation defect (Supplementary Fig. 6c, d). Together, these findings further support our hypothesis that ribosomal haploinsufficiency caused by loss of RPG is responsible for the observed translation defects. Finally, we analyzed the amount of total RNA in monosomic cells compared to the parental control. Since rRNA represents approximately 80% of all RNA in eukaryotic

cells[31], the total RNA amount can serve as a proxy for rRNA abundance and ribosome content[32]. Strikingly, RM 10 and RM 13 comprised 70–80% of the total RNA abundance of the parental control (Fig. 5k). We propose that a loss of one gene copy coding for ribosomal proteins is responsible for the reduced translation in monosomic cells due to impaired ribosome biogenesis.

**Loss of p53 is essential for the proliferation and viability of monosomies.** To determine how *TP53* affects the cellular response to monosomy, we restored the p53 expression in monosomic cell lines using doxycycline inducible expression system (Fig. 6a and Supplementary Table 1). The new cell lines were labeled with ip53 (e.g., RM13 ip53). We then titrated doxycycline in each cell line to determine the doxycycline concentration that restores the p53 abundance to the levels observed in the wild type RPE1 p53$^{+/+}$ cell line (Supplementary Fig. 7a). The expression of p21, a downstream target of p53, was similar as

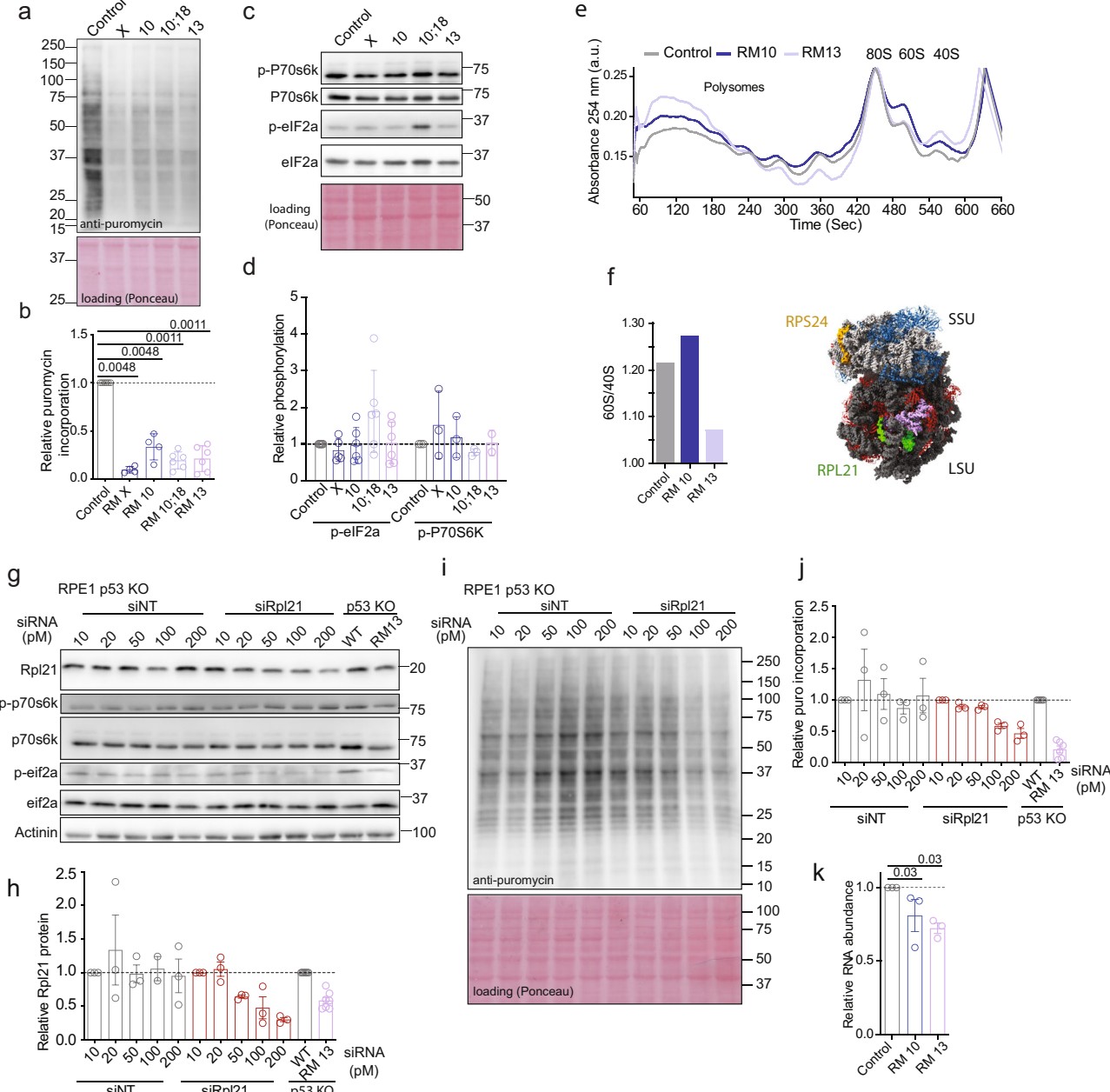

**Fig. 5 Ribosome and translation defects in monosomies. a** Evaluation of protein synthesis rates in monosomies. Equal amounts of puromycin-labeled cell lysates were immunoblotted and analyzed using anti-puromycin antibody. Ponceau staining was used as a loading control. **b** Quantification of mean puromycin intensities from immunoblotting. The intensities were normalized to Ponceau staining. Bars display the mean ± SEM of four independent experiments for RM X and RM 10, and six independent experiments for other cell lines. Statistical analysis was performed by comparing to the respective controls using non parametric Matt Whitney two tailed T-test; p-values are shown on the plot. **c** Immunoblotting of mTOR target p70S6k and phospho-p70S6K and the integrated stress response marker eIF2α and phospho-eIF2α. **d** Quantification of relative phosphorylation levels of P70S6K and eIF2α normalized to total protein levels, respectively. Bar graphs display the mean ± SEM of three independent experiments for P70S6K and six for eIF2α. **e** Polysome profiles obtained from diploid and monosomic cell lines. Profiles are representative of two independent experiments. The profiles were adjusted to the 80S peak. **f** Quantification of 60S/40S ratio of the representative polysome profiles. Peak absorbance for 60S and 40S were used for the calculations. Ribosome model shows the location of RPS24 (yellow) and RPL21 (green) proteins within the small and large ribosome subunit, respectively. **g** siRNA mediated titrated knockdown of RPL21 in RPE1 WT p53 KO cell line. siNT was used as a control; RPE1 p53 KO WT and RM 13 are shown as a non-transfected control. The western blots are representative of three independent experiments. **h** Quantification of RPL21 knock down efficiency. siNT and siRPL21 values were normalized to respective 10 pm samples. Bars display the mean ± SEM of three independent experiments. **i** The effect of siRNA mediated knockdown of RPL21 on protein synthesis. Equal amounts of protein lysates from siNT and siRPL21 samples were immunoblotted against anti-puromycin antibody. Ponceau staining was used as a loading control. The western blots are representative of three independent experiments. **j** Quantification of translation rate after RPL21 knockdown. siNT and siRPL21 values were normalized to respective 10 pm samples. Bars display the mean ± SEM of three independent experiments. **k** Measurement of total RNA as a surrogate for rRNA levels and ribosome content. Bar graphs display the mean ± SEM of three independent experiments. Statistical analysis was performed by comparing to the respective controls using non parametric Matt Whitney two-tailed T-test; p-values are shown on the plot. Source data are provided in a Source Data file.

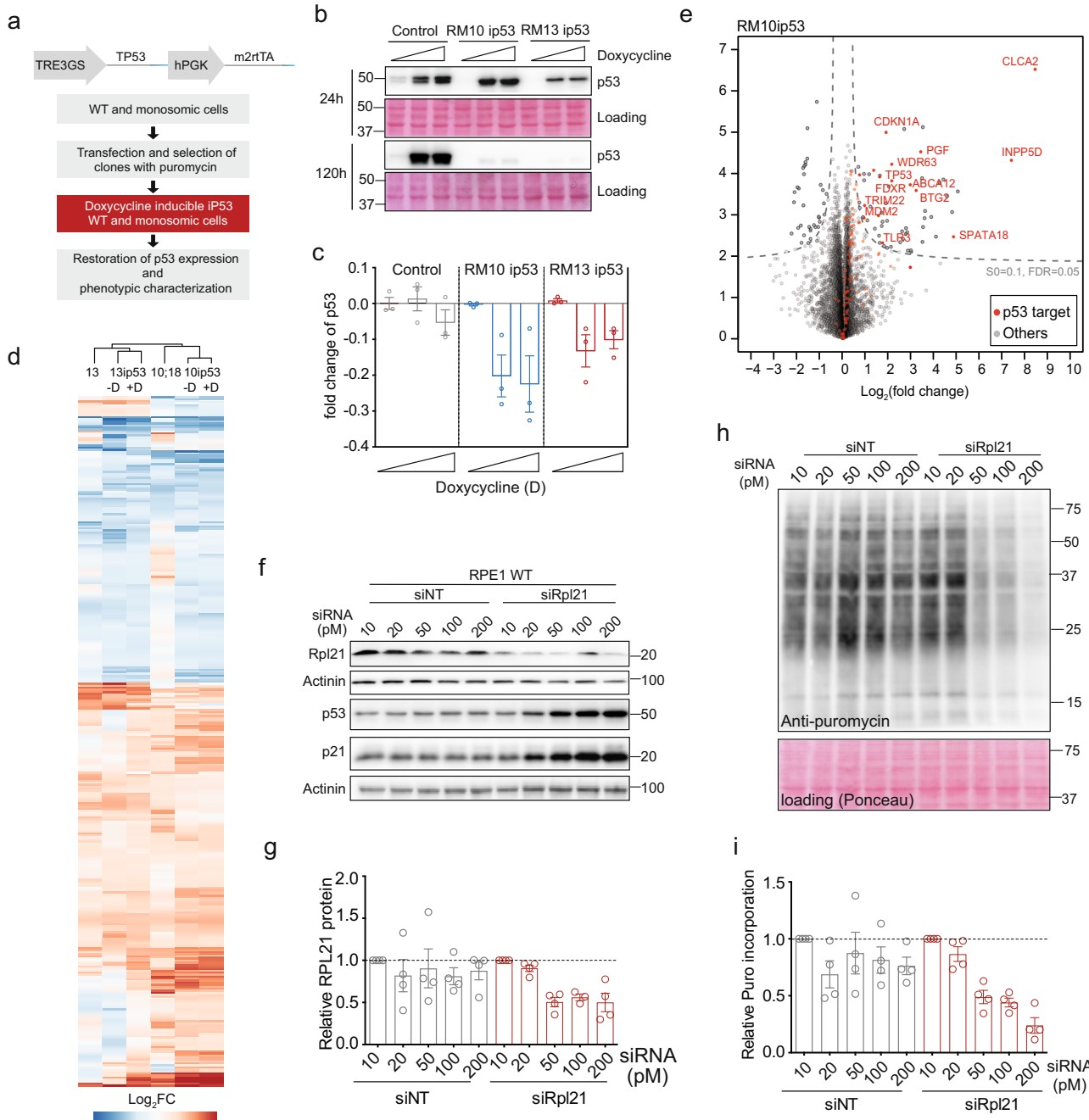

**Fig. 6 Loss of p53 is essential for proliferation of monosomies. a** Schematics depicting the construction of monosomic cell lines with doxycycline inducible p53 expression. **b** Immunoblotting of p53 after induction with doxycycline for 24 and 120 h. Ponceau staining was used as a loading control. **c** Quantification of the p53 intensities from **b**. The plots display the mean differential expression of p53 at 24 and 120 h of doxycycline treatment. All values were normalized to the respective loading control. D denotes doxycycline. Bars display the mean ± SEM of three independent experiments. **d** Heat map depicts the differentially regulated mRNA expression in monosomies compared to diploids, Hierarchical clustering of Euclidean distance. Gene expression fold changes greater than 2 or less than −2 are used for heat map. **e** Volcano plot showing the transcripts of p53 targets upregulated in RM10 ip53 cell line with doxycycline treatment compared to no doxycycline (log2FC). Red dots represent known p53 targets. **f** siRNA mediated titrated knockdown of RPL21 in RPE1 induces p53 expression in a concentration-dependent manner. siNT was used as control. RPE1 p53 KO WT and RM 13 were used as a non-transfection control. The western blots are representative of three independent experiments. **g** Quantification of relative RPL21 levels from immunoblotting. All values were normalized to respective 10 pm samples. Actinin served as a loading control. Bars display the mean ± SEM of three independent experiments. **h** siRNA mediated knockdown of RPL21 reduces translation rate in RPE1 cells. The protein lysates from siNT and siRPL21 puromycin-labeled samples were immunoblotted with an anti-puromycin antibody. Ponceau staining was used as a loading control. The western blots are representative of three independent experiments. **i** Quantification of relative puromycin incorporation from immunoblotting. All values were normalized to respective 10 pm samples. Ponceau served as a loading control. Bars display the mean ± SEM of three independent experiments. Source data are provided in a Source Data file.

in the p53 proficient RPE1 (Supplementary Fig. 7a). Having these cells at hand, we restored the p53 expression to the wild type levels and analyzed its impact on monosomic cells. Strikingly, we found that the expression levels of p53 quickly decreased in monosomic cells treated with doxycycline (Fig. 6b, c). Microscopy of ip53 cell lines revealed that the cells expressing p53 were outgrown by the cells without p53 in monosomies (Supplementary Fig. 7b–d). These findings demonstrate that monosomy is not compatible with functional p53 pathway in human cells.

Next, we analyzed transcriptome changes in monosomic cells with restored p53 expression by RNAseq of RPE1 ip53, RM10 ip53, and RM13 ip53 after 48 h of doxycycline treatment. Functional p53 does not alter the global response to monosomy, as most deregulations remained similar with restored p53 expression (Fig. 6d and Supplementary Data 4). However, we identified a significant upregulation of 12 known downstream targets of p53 (Fig. 6e and Supplementary Data 4), among which p21 (CDKN1A) and BTG2 are the key pro-survival factors and early responders to p53 activation that mediate the cell cycle arrest. Accordingly, analysis of the cell cycle progression revealed an increased accumulation of G1 cells 24 h after restored TP53 expression (Supplementary Fig. 7e). We conclude that monosomic cells with restored p53 activate this pathway, which subsequently triggers the G1 cell cycle arrest and blocks proliferation.

Defective ribosome biogenesis is a known trigger for p53 activation[33]. Since the monosomies suffer from impaired ribosome biogenesis, we hypothesized that ribosomal haploinsufficiency may be responsible for the p53 activation. Depletion of RPL21 in diploid p53-positive cells to the levels similar as in RM13 (Fig. 6f, g) reduced the translational efficiency (Fig. 6h, i). Importantly, we observed that already a minor reduction of Rpl21 abundance resulted in a robust accumulation of p53 and p21 (Fig. 6f). Thus, a partial depletion of a single ribosomal protein to levels resembling its abundance in monosomic cells is sufficient to induce a robust p53 activation. This suggests that the incompatibility of functional p53 and monosomy is due to haploinsufficiency of the ribosomal genes.

**Loss of p53 and ribosome deprivation are hallmarks of monosomic cancers.** One prediction from the above described observations is that cancers with monosomy show reduced expression of ribosomal genes and enriched p53 pathway mutations. To test this, we analyzed transcriptomes from the Cancer Cell Line Encyclopedia (CCLE)[34]. Cell lines were categorized into Monosomy and Disomy (see "Methods" section). By single sample gene set enrichment analysis (ssGSEA)[35], we show that ribosome-related pathways are strikingly downregulated in Monosomy cancer cell lines compared to Disomy ones (Fig. 7a and Supplementary Fig. 8a and Supplementary Data 5, 6). Additionally, we found a significant downregulation of p53 signaling pathways (Fig. 7a). We then filtered the Monosomy category to two groups: a group with RPG expression lower than the median of the entire cohort, and a group with a higher RPG expression. Comparison of these groups revealed a significantly reduced TP53 pathway activity in cells with reduced RPG abundance (Supplementary Fig. 8b). Next, we examined the relationship between p53 loss and monosomy in-vivo. We stratified the Cancer Genome Atlas (TCGA) pan-cancer samples into Monosomy, Disomy and Polysomy (see "Methods" section). We then calculated the "p53 classifier score" that quantifies the extent of phenocopying TP53 loss in individual samples[36]. Of the 15 cancer types, Monosomy shows significantly higher TP53 scores compared to Disomy in 13 cancer types, while Polysomy only in three cancer types (Fig. 7b). Previous studies showed that tumors

with high SCNA are enriched for TP53 mutations[37]. To evaluate whether the increased TP53 classifier score in Monosomy might be a confounding effect of increased SCNA in this category, we measured the aneuploidy score of Monosomy and Polysomy tumors and compared to TP53 mutations. While the aneuploidy score defining the SCNA levels is comparable in Monosomy and Polysomy, the TP53 mutations are enriched in Monosomy (Fig. 7c). Based on these results we propose that ribosome biogenesis defect is common in monosomic cancers due to the haploinsufficiency of ribosomal genes and that, therefore, monosomy is incompatible with functional p53 pathway.

## Discussion

Recent years have witnessed a breakthrough in understanding the cellular consequences of chromosome gains, but examination of chromosome losses has been difficult. We were able to analyze seven human monosomies of chromosomes 7, 10, 13, 18, 19p, and X. While this is a limited number of monosomies, it provides an insight to the consequences of monosomy in human cells. We demonstrate that monosomy leads to impaired proliferation, genomic instability, and deregulated gene expression. All analyzed monosomies showed reduced expression of RPGs and impaired translation. We also show that loss of p53 function is a precondition for proliferation of monosomic cells. Our findings are further supported by computational analysis of CCLE and TCGA data, which revealed that monosomy in cancer cells correlates with reduced ribosomal functions and dysfunctional p53 pathway.

**Monosomy induces genomic instability and impairs proliferation.** Most of our understanding of cellular response to aneuploidy comes from the analysis of trisomy that impairs proliferation and induces genotoxic and proteotoxic stress. Monosomy also impairs cell proliferation, but the underlying molecular mechanisms likely differ from the effects of trisomies, since no proteotoxic stress was observed in response to monosomy. The impaired proliferation of monosomic cells might be either due to haploinsufficiency of genes required for proliferation, or due to the fact that spontaneously arising deleterious recessive mutations can no longer be compensated by the second gene copy. We observed an increased chromosome missegregation and mitotic delay in several monosomic cell lines, while increased accumulation of DNA damage was observed in three monosomic cell lines. Intriguingly, the defects were generally milder in cells that were rendered p53 deficient by shRNA expression. Moreover, monosomy of chr. X showed no increase in genomic instability. This suggests that the changes in maintenance of genome stability are not a uniform outcome of monosomy. Genomic instability in trisomic cells is associated with replication stress[3,22,38]; however, none of these phenotypes were observed in monosomies. The deregulation of proteins linked to DNA replication in trisomic cells likely depends on TP53—while p53 proficient trisomic cell lines downregulate DNA replication factors[3,22,38], the p53 deficient trisomic cell lines analyzed so far upregulated these factors[39]. Whether the absence of replication stress in monosomies is due to the p53 loss is difficult to address because of the toxicity of p53 in monosomies. Thus, imbalanced chromosomal content, either due to a loss or a gain of chromosome, is detrimental for proliferation and for maintenance of genome stability in human cells, but the underlying molecular causes likely differ in monosomic and trisomic cells.

**Gene expression changes induced by monosomy.** Only little is known about gene expression changes due to monosomy. In a

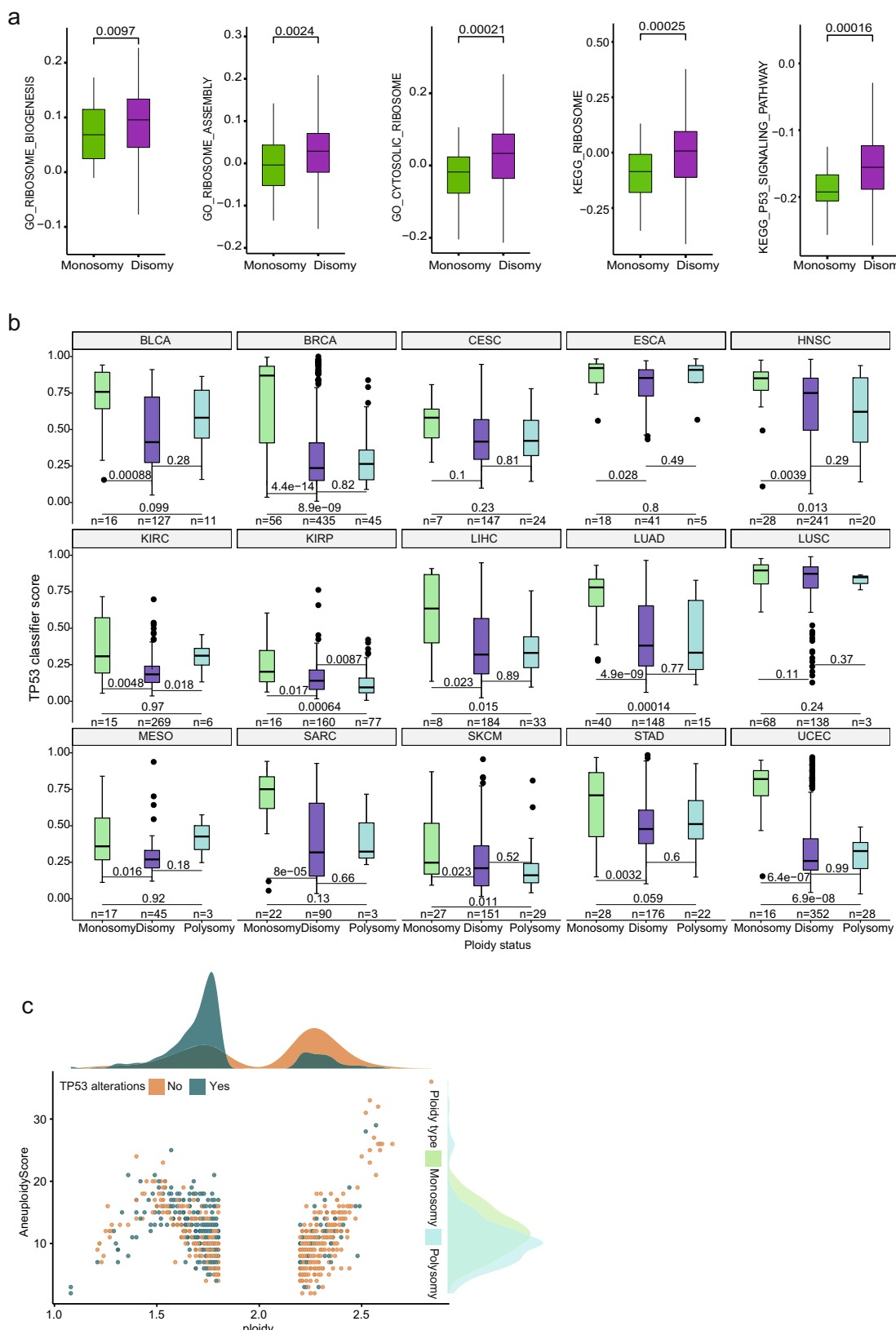

recent transcriptome analysis of human monosomic blastocysts, altered transcript abundances were observed for several hundreds of genes located on the affected monosome as well as genome wide, but the study was hampered by low numbers of analyzed transcripts, restrictions to embryonal cells and genetic diversity of the unrelated embryos[40]. Our systematic transcriptome and proteome analysis of monosomic cell lines compared to their parental cell line showed that the expression of the genes located on the monosome is indeed reduced, but the expected lower levels were observed only for 20% of monosomically encoded proteins. Both transcriptional and posttranscriptional mechanisms contribute to this abundance changes. Study of gene dosage effects

**Fig. 7 TP53 loss and impaired ribosome biogenesis are the hallmarks of monosomic cancers. a** Transcriptome analysis revealed the downregulation "Ribosome biogenesis", "Ribosome assembly" and "p53 pathway" in monosomic ($n = 48$) compared to disomic cell lines ($n = 349$) in CCLE datasets. The *y*-axis shows the ssGSEA derived enrichment scores for gene ontology (GO) terms "Ribosome biogenesis", "Ribosome assembly" and "Cytosolic ribosome" and KEGG pathways "Ribosome" and "p53 signaling", respectively. All *p*-values are based on one-sided Wilcoxon rank sum tests. All box plots include the median line, the box denotes the interquartile range (IQR), and whiskers denote the rest of the data distribution. **b** TP53 classifier score estimating the extent of phenocopying *TP53* loss in monosomy compared to diploid and polysomy tumors across pan cancer TCGA dataset. Number of tumors in each group is shown on the plot. Statistical significance was calculated using one-sided Wilcoxon rank sum test. All box plots include the median line, the box denotes the interquartile range (IQR), whiskers denote the rest of the data distribution and outliers are denoted by points greater than ±1.5 × IQR. Cancer types are annotated as follows: BLCA bladder urothelial carcinoma, BRCA breast invasive carcinoma, CESC cervical squamous cell carcinoma, ESCA esophageal carcinoma, HNSC head and neck squamous cell carcinoma, KIRC kidney clear cell carcinoma, KIRP kidney renal papillary cell carcinoma, LIHC liver hepatocellular carcinoma, LUAD lung adenocarcinoma, LUSC lung squamous cell carcinoma, MESO mesothelioma, SARC sarcoma, SKCM skin cutaneous melanoma, STAD stomach adenocarcinoma, UCEC uterine corpus endometrioid carcinoma. **c** Scatter plot showing the ploidy distribution. Samples with ploidy ranging from 1.80 to 2.19 are defined as disomic, 1.66–1.90 as Monosomy and 2.0–2.27 as polysomy. The top marginal density histogram shows that TP53 alterations. The right marginal density histogram depicts the aneuploidy score in monosomy and polysomy.

---

upon partial autosomal deletions in drosophila cell line S2 suggested gene-specific mechanism affecting mRNA levels[41]. We observed that in human monosomies, the major gene expression adjustments occur on protein level. We envision two possible scenarios: first, the translation of mRNAs originating from genes encoded on monosomic chromosome might be selectively increased, or second, the protein degradation becomes reduced. Our findings indicate that cells utilize multiple routes to alleviate the consequences of gene expression changes.

Monosomy alters not only the expression of the genes encoded on monosomic chromosomes, but also affects the genome-wide expression landscape. The differentially regulated pathways revealed only limited overlap with our previous analysis of trisomic cells. It should be noted, however, that the trisomic cells were p53 proficient, while the monosome are p53 negative. Comparisons of monosomies with ip53 suggest rather that at least in monosomic cell, the p53 pathway has only mild effect on differential gene expression. Additionally, the deregulated genes and pathways largely differ among the monosomic cells lines. Thus, the identity of individual genes located on individual monosomes dominates over the shared consequences of chromosome loss.

**Chromosome loss impairs ribosome biogenesis**. All analyzed monosomies downregulated cytoplasmic ribosomes and translation. Ribosome biogenesis and assembly are highly complex processes involving transcription, modification, and processing of precursor rRNAs, synthesis, import, and export of ribosomal subunits. Even subtle shifts in the availability of ribosomes, such as those created by ribosome haploinsufficiency, impairs cellular growth, and proliferation[42]. Every human chromosome, with exception of chromosome 7, carries at least one gene that codes for ribosomal proteins or an rDNA cluster[30]. This is also true for chromosome X and, importantly, one of the genes that escapes dosage compensation codes for ribosomal protein *RPS4X*. It has been proposed that haploinsufficiency of *RPS4X* contributes to the pathophysiology of monosomy X, or Turner syndrome[43], although the importance remains debated. Other pathways known to affect translation efficiency, such as integrated stress response[29,44] or mTOR pathway[28], were not changed in response to monosomy. Therefore, we propose that impaired ribosome biogenesis caused by the haploinsufficiency of ribosomal protein-coding genes as the common consequence of monosomy. An ultimate test for our hypothesis would be a rescue of the translation defect in monosomic cells by restoring the levels of ribosomal proteins. Our attempts to rescue the RPL21 expression were not successful, as the excessive protein was readily degraded by proteasome. This is likely due to tight regulation of RPG expression that renders individual ribosome subunits generally

resistant to overexpression[45,46]. Novel approaches will have to be developed to perform this experiment in future.

**Loss of p53 is a precondition of monosomy**. Autosomal monosomy appears to be incompatible with functional p53 pathway. Our original attempts to obtain monosomic cells in p53 proficient cell lines were unsuccessful. Previously published CRISPR/Cas9-based technique was efficient only to generate monosomy X[47]. Monosomy of 3p was engineered with CRISPR/Cas9 system in lung cells immortalized with the SV40 large T antigen, which perturbs the retinoblastoma protein and p53 tumor suppressor proteins[16]. Previous observations also revealed anti-correlation between chromosome loss and functional p53 in patients with MDS, where *TP53* mutation was associated with chromosome 5q loss in 47% of patients, while only 1.5% (4/263) of patients without 5q aberrations carried *TP53* mutations[48]. Tumors in zebrafish with ribosomal haploinsufficiency also require a loss of p53 expression[49]. Together, the data suggest an important role of the p53 pathway in the viability of monosomic cells. Restoring the p53 function in monosomic cells gave us the unique opportunity to evaluate the function of p53. While the global cellular response to monosomy was not strikingly affected, we found expression changes in a small subset of factors enriched for direct p53 targets such as *CDKN1A*, *BTG2*, *FDXR*, *SPATA18*, *CLCA2*, and others. Several of the induced factors, for example *CDKN1A* (p21) and *BTG2* regulate G1/S transition of the cell cycle or play an important role in cellular response to stress conditions (*BTG2*, *FDXR*). Computational analysis of cancer genomes confirms the principal incompatibility of functional p53 with monosomy, as well as general downregulation of ribosomal pathways. We propose that the defect in ribosome biogenesis may activate the p53 pathway in monosomic cells. In addition, increased genomic instability may contribute to p53 activation. Our findings provide insight into how chromosome loss affects gene expression landscape in human cells, and offer rationale for why monosomic tumors often harbor mutations in the p53 pathway.

## Methods
**Cell lines and treatments**. Human retinal pigment epithelium cell line RPE1 immortalized with hTERT overexpression was used. RPE-1 cells were cultured in DMEM + GlutaMAX^{TM}-I medium supplemented with 10% Fetal Bovine Serum and Pen strep (Gibco). RM 13 KO cell line was generated by pre-designed zinc finger nucleases (Sigma) as described previously[50]. RM X, RM 10, RM 10;18, and RPE1 p53 KO cell lines were generated by CRISPR/Cas9-based targeting of the *TP53* gene. For these, the gRNA against *TP53* gene was cloned in pX330 vector (Addgene: 42230) according to a modified protocol from[51] and used to transfect RPE WT cells. Single cells clones were tested for the loss of p53 expression by sensitivity to Nutlin and immunoblotting for p53 and p21. To assess the copy number status, single cell derived clones were subjected to low-pass whole genome sequencing. Chromosome loss was further validated by chromosome painting. The

monosomic clones (KD) were derived from RPE1 p53 shRNA cell line via induction of chromosome missegregation by combined low dose of the MPS1 inhibitor (NMS-P715) with an allosteric inhibitor of the CENP-E kinesin (GSK923295) as described in ref. [2], and were a kind gift from Dr. Rene Medema (Netherlands Cancer Institute, The Netherlands).

For p53 rescue experiments, RM 10;18 and RM 13 cell lines were transduced with virus generated from an all-in-one tetracycline inducible plasmid with p53 (pMOV T11 p53) expressed under tet promoter[52], selected in the presence of 2 µg/mL puromycin for 72 h and subsequently maintained in the presence of 0.5 µg/mL puromycin. RPE1 WT cells were transduced with *TP53* cloned into doxycycline inducible system (Tet on system, Clonetech) and selected with blasticidine (1 µg/mL) and puromycin (2 µg/mL). All lines were grown at 37 °C in a humidified 5% $CO_2$ incubator. All cell lines were tested for mycoplasma contamination. To minimize the occurrence of secondary genomic changes, original stocks were thawed for every experiment and maintained for maximum of 4–5 passages. The used cell lines are listed in Supplementary Table 1.

**Lentivirus transfection and transduction**. The pMOV T11 (kind gift of Dr. Bernhard Schiedlmeier, Medizinische Hochschule Hannover, Germany) and pRetroX-TRE3G and pRetroX-Tet3G vectors containing the *TP53* coding sequence were mixed with pMDLg/pRRE (gift from Didier Trono, Addgene plasmid #12251) and pMD2.G (gift from Didier Trono, Addgene plasmid #12259) lentiviral plasmids and transfected to 80% confluent HEK293T cells using Lipofectamine 2000 (Thermo Fischer Scientific) according to the manufacturer's instructions. On the next day, the medium with transfection reagent was replaced with fresh medium. Forty-eight hours of post-medium change, the viral supernatant was collected.

RPE1 p53 KO and RM 10;18 and RM 13 were transduced with viral supernatant supplemented with 5 mg/mL polybrene and incubated 12–16 h at 37 °C in a humidified 5% $CO_2$ incubator. After 48 h, the medium was replaced with selection medium containing the respective antibiotics.

**siRNA transfection**. For the knockdown of RPL21 using siRNA, $5 \times 10^5$ were plated on 6 cm dishes on the day before transfection. Different concentrations (10, 20, 50, 100, and 200 pm) of siNT (control) and siRPl21 were used. The transfection of siRNA was performed with Lipofectamine 2000 according to the manufacturer's instructions. Seventy-two hours of post-transfection, cells were collected for immunoblotting to verify the RPL21 knockdown efficiency. The sequences of siRNAs are listed in Supplementary Table 2.

**Metaphase spread and chromosomal painting**. Cells were grown to 70–80% confluence, treated with 400 ng/mL colchicine for 5–6 h, collected by trypsinization, and centrifuged at 500 × *g* for 10 min. Cell pellets were resuspended in 75 mM KCl and incubated for 10–15 min at 37 °C. Cells were pelleted at 500 g for 10 min and suspended in 3:1 methanol/acetic acid to fix the cells, washed several times in 3:1 methanol/acetic acid. Fixed cells were dropped on a glass slide and dried at room temperature for 15 min. Each sample was labeled with chromosome FISH probes (Chrombios) specific for a monosomic chromosome and a control chromosome as per manufacturer's instructions. Briefly, chromosome spreads were incubated with probe mixture (1 µL of each probe, adjusted to 10 µL with HybMix buffer). After denaturation at 72 °C for 6 min, slides were kept at 37 °C in a humid chamber overnight. Slides were washed for 5 min in 2× saline sodium citrate (SSC) solution and then for 1 min in prewarmed 70 °C 0.4× SSC, 0.1% Tween solution, and, finally, in 4× SSC, 0.1% Tween solution for 5 min at room temperature. Then slides were incubated for 30 min at 37 °C with 100 µL fluorescein isothiocyanate (FITC) mouse anti-digoxin (Jackson Immuno Research) solution (1:300 in 4× SSC/ 0.1% Tween) and washed twice in 45 °C pre-warmed 4× SSC/0.1% Tween solution for 5–10 min. Finally, cells were stained with DAPI and microscopic analysis was carried out using 3i software and spinning disc confocal microscopy (see below). For each sample, at least 25 metaphases were captured and analyzed.

**Immunofluorescence staining and microscopy**. RPE1 p53 KO and monosomic cells ($1 \times 10^4$ cells) were plated in black 96-well glass-bottom plates and grown in DMEM to the desired confluence. Cells were fixed with freshly prepared 3.7% formaldehyde for 15 min at RT and permeabilized with 0.5% Triton X-100 in PBS for 5 min. For blocking, cells were incubated with 3% BSA and 0.1% Triton X-100 for 30 min at RT and stained with anti-gamma H2AX (Abcam 2893), anti CENP B (Santa Cruz, sc376392), anti p53 (Santa Cruz, sc126) overnight at 4 °C in humidified chamber. Next day, the primary antibody was washed off and incubated with secondary antibody at room temperature in dark for 1 h. After secondary antibody was washed off, the nuclei was counter stained with Sytox green and DAPI. For micronuclei and anaphase bridge quantification, the cells were fixed as above and counter stained with DAPI and Sytox green.

Spinning disc confocal laser microscopy was performed using a fully automated Zeiss inverted microscope (AxioObserver Z1) equipped with a MS-2000 stage (Applied Scientific Instrumentation, Eugene, OR), the CSU-X1 spinning disk confocal head (Yokogawa) and LaserStack Launch with selectable laser lines (Intelligent Imaging Innovations, Denver, CO). Image acquisition was performed using a CoolSnap HQ camera (Roper Scientific) and a 20×-air (Plan Neofluar × 40/

0.75, Plan Neofluar × 20/0.75) under the control of the SlideBook 6 × 64 program (SlideBook Software, Intelligent Imaging Innovations, Denver, CO, USA).

**Live cell imaging**. For the live cell imaging, cells expressing H2B-Dendra2 were seeded in a 96-well plate at $1 \times 10^4$ cells per well in standard cell culture medium. Cells lacking the fluorescent tag to visualize DNA were labeled with Hoechst 33342 (1 µg/mL). The medium was replaced with FluoroBrite medium before live-cell imaging. Imaging was performed using an inverted Zeiss Observer Z1 microscope (Visitron Systems) equipped with a humidified chamber (EMBLEM) at 37 °C, 40% humidity, and 5% $CO_2$ using CoolSNAP HQ2 camera (Photometrics) and X-Cite 120 Series lamp (EXFO) and Plan Neofluar ×20, or ×10 magnification air objective NA 1.0 (Zeiss, Jena, Germany). Cells were imaged for 24 h with 8 min time-lapse. Images were analyzed using Slidebook (Intelligent Imaging Innovations, Inc., Goettingen, Germany) and ImageJ (National Institutes of Health). To determine the time spent in mitosis, the period from nuclear envelope breakdown (NEBD) until end of anaphase was quantified.

**Cell proliferation and soft agar colony forming assay**. For proliferation assay, cells were seeded in triplicates into the wells of a 96-well plate ($1.5 \times 10^3$ cells/well). In total five plates, one for each day, were prepared. To measure the proliferation, Cell Titer-Glo (Promega) was used according to the manufacturer's instructions. All the measurements were normalized to Day 0. For soft agar assay, 1% low melting agarose combined with an equal volume of DMEM was added to 12 well dish as a bottom layer. Subsequently, 0.7% low melting agarose was mixed with an equal volume of cell suspension containing 1000 cells and immediately layered on solidified agar base in duplicates per cell line. The wells were then filled with medium containing 10% FBS and 5% Pen-Strep. Medium was replaced every 3 days and the colonies were counted after 3 weeks. Each well was divided into eleven fields and colonies in each field were counted using an inverted microscope (Motic AE2000).

**Immunoblotting**. Whole-cell lysates were obtained using RIPA buffer supplemented with protease and phosphatase inhibitors (Roche). An amount of 10 µg of protein was then resolved on 10–12.5% polyacrylamide gels and transferred to nitrocellulose membranes using the semi-dry technique. Ponceau staining was performed by incubating the membrane for 5 min in Ponceau S solution (2 (w/v) in 1% (v/v) acetic acid). After blocking in low fat, 5% milk in Tris-Buffered Saline with Tween 20 for 1 h at room temperature, membranes were incubated overnight at 4 °C with the primary antibodies. Antibodies used in this study are listed in Supplementary Table 2. After incubation with horseradish peroxidase-conjugated secondary antibodies, horseradish peroxidase substrate was added and luminescent signals were quantified using an Azure c500. Protein bands were quantified using ImageJ software. For the normalization of western blotting results, we used housekeeping gene α-actinin or Ponceau staining as indicated in the figure legends. For Ponceau-based normalization, we used a large region between 35 and 60 kDa that contains several bands for normalization.

**RNA extraction and RT-PCR**. mRNA was extracted using Qiagen RNeasy mini kit as per the manufacturer's instructions. Two microgram of genomic DNA-free mRNA was used for cDNA synthesis. cDNA was synthesized using iScript™ Advanced cDNA Synthesis Kit as per the instructions. As a control for cDNA synthesis efficiency, RNA spike (TATAA universal RNA spike I) was added in equal amounts into master mix of cDNA. cDNA was diluted 1:10 before using it for qPCR analysis. For qPCR, SYBR green based assay was used from Biorad (Sso advanced universal SYBR Green). The Ct values for the gene of interest is normalized either to RPL27 or Spike. The primers used in this study are listed in Supplementary Table 3.

**Cell cycle analysis**. Cell cycle analysis was performed by labeling the replicating cells with EdU (5-ethynyl-2′-deoxyuridine) and DNA with DAPI (4′,6-diamidino-2-phenylindole). Briefly, cells were cultured as described above and EdU was added 30 min before harvesting the cells. Subsequently, for EdU detection, the cells were fixed and permeabilized for 15 min with Fix perm (Thermo Fisher scientific), followed by incubation with EdU Click-iT cocktail (Invitrogen) as per the manufacturer's instructions. Cells were resuspended in PBS containing RNase (10 µg/mL) and DAPI and measured using Attune Nxt acoustic focusing cytometer (Life Technologies, Carlsbad, USA) and Attune NxT software v3.1.1243.0. The gating and further details of the analysis are in Supplementary Fig. 9.

**Puromycin labeling to determine translation rate**. To determine the translation rate, $1.5 \times 10^6$ cells were plated in 10 cm dish on the day before the puromycin labeling. Next day, cells should be actively growing with desired confluence of 70–80%. For labeling, 10 µM of puromycin was added directly to cell culture dish, mixed well, and placed in incubator at 37 °C for 15 min. Puromycin was washed off with PBS and the cells were collected for protein extraction and immunoblotting as described above. Equal amounts of protein lysates were loaded on 12.5% acrylamide gel and the puromycin incorporated nascent peptides were identified using

anti puromycin antibody. The intensity of puromycin was normalized to ponceau, which served as loading control.

**17-AAG sensitivity**. To analyze the sensitivity to protein folding inhibitor 17-AAG (inhibitor of HSP90), $1.5 \times 10^3$ were plated in triplicates on white 96-well glass bottomed plate. On the following day, the medium was replaced with fresh medium containing 17-AAG at desired concentrations or DMSO. Immediately, cells were placed in incubator with 5% $CO_2$ at 37 °C. After 72 h, the cell viability was measured using Cell Titer-Glo according to manufacturer's instructions. All the values were normalized to DMSO control.

**Polysome profiling**. Polysome profiling was performed as in ref. [53]. Cells were grown in 15 cm dishes (80% confluency at the time of experiment) and ten dishes were used for each cell line. Cells were treated with 100 µg/mL cycloheximide (CHX) for 10 min and collected immediately by gently scraping with ice cold CHX/PBS. Cells were pelleted and flash frozen using liquid nitrogen and stored at −80 °C until further use. Cells were lysed using ice cold low salt lysis buffer (50 mM KCL, 20 mM Tris HCL, pH 7.4, 10 mM $MgCl_2$, 1% Triton X-100, 1 mM DTT, 0.5% sodium deoxycholate, 100 µg/mL cycloheximide, RNAse inhibitiors, protease, and phosphatase inhibitors) and incubated on ice for 10 min. The lysate was centrifuged at 2000×g for 5 min to pellet nuclei and large debris. The supernatant was transferred to a fresh tube and centrifuged at 13,000 × g for 5 min and the supernatant was transferred to a fresh tube. In the meantime, a linear sucrose gradient (7–47%) was prepared using Biocomp Gradient Master 108. The RNA concentration of cleared lysate was measured using nanodrop and equal amounts of lysate was layered on the top of gradient and centrifuged at 260,110 × g on SW41 rotor for 90 min at 4 °C. The UV absorbance of the gradients were measured starting at the bottom of the gradient using Bio Rad BioLogic FPLC system and BioLogic Optics Module OM-10.

The raw polysome profiles were smoothed using a Savitzky-Golay filter with a window size of 51–61 data points and a third order polynome in a Python script. For normalization by 80S maximum peak intensity, the data points of each monosomic cell line profile was multiplied by the ratio of its 80S maximum peak intensity to the wild types 80S maximum peak intensity. Based on the normalized profiles, the ratios of 80S, 60S, and 40S maximum peak intensities between the wild type and monosomies were calculated. Ribosome structure presentation is based on the human ribosome (PDB identifier 6Y2L, available via). The model shows the structure of the human ribosome in the post-translocation state during elongation. The model was built on cryo-EM data with 3.00 Å resolution with the UCSF ChimeraX software[54]. We chose this structure to display the ribosomal proteins important in context of this work.

**DNA libraries**. Genomic DNA was extracted from the cells using the DNA Blood Mini kit (Qiagen). Library preparation was performed with a Beckman Biomek FX automated liquid handling system, with 500 ng starting material using SPRIworks HT chemistry (Beckman Coulter). Samples were prepared with custom six base pair barcodes to enable pooling. Library quantification and quality control was performed using a Fragment Analyzer (Advanced Analytics Technologies, Ames, USA).

**Genomic sequencing and analysis**. WGS was pursued on an Illumina HiSeq 2500 platform (Illumina, San Diego, USA), using 50 basepair single reads for low-pass sequencing. For all samples the GC-normalized data was aligned against the Genome Reference Consortium Human Build 38 patch release 13 (GRCh38.p13) using the intersect function of bedtools2 (version 2.29.1) to map the known genes to the measured coverages. The values were converted into log2 data and the median coverage per read was calculated for all known genes with at least one mapped coverage. To shift the values around zero, the median coverage of each cell line was subtracted for all values, resulting in a normalized population centered on 0.

**RNA-seq library preparation and sequencing**. NGS-Sequencing and library preparation was conducted at the NGS- Integrative Genomics Core Unit (NIG), Institute of Human Genetics, University Medical Center Göttingen (UMG).

Quality and integrity of RNA was assessed with the fragment analyzer from Advanced Analytical by using the standard sensitivity RNA analysis Kit (DNF-471). All samples selected for sequencing exhibited an RNA integrity number over 8. RNA-seq libraries were performed using 200 ng total RNA of a non-stranded RNA Seq, massively-parallel mRNA sequencing approach from Illumina (TruSeq stranded total RNA Library Preparation, Illumina). Libraries were prepared on the automation (Beckman Coulter's Biomek FXP workstation). For accurate quantitation of cDNA libraries a fluorometric based system, the QuantiFluor™dsDNA System from Promega were used. The size of final cDNA libraries was determined by using the dsDNA 905 Reagent Kit (Fragment Analyzer from Advanced Bioanalytical) exhibiting a sizing of 300 bp in average. Libraries were pooled and sequenced on the Illumina HiSeq 4000 (SE; 1 × 50 bp; 30–35 Mio reads/sample). Sequence images were transformed with Illumina software BaseCaller to BCL files, which was demultiplexed to fastq files with bcl2fastq v2.20.0.422. The quality check was done using FastQC.

**Mapping and normalization and analysis**. Sequences were aligned to the genome reference hg38 sequence using the STAR aligner version 2.5.2a[55]. Subsequently, read counting was performed using featureCounts[56] (version 1.5.0-p1). Read counts were analyzed in the R/Bioconductor environment (version 3.6.1, www.bioconductor.org) using the DESeq2 package version 1.14.1[57]. For further analysis, all counts were normalized by shifting the replicates to the same median, which was calculated without the monosomic genes to adjust the samples for the loss of a chromosome and the subsequent lower gene expression. For all monosomic cell lines, the log2 median intensity of three replicates was calculated and the log2 median intensity of three replicates of the wild type parental cell line was subtracted to calculate comparable fold changes.

**Preparation of tandem mass tag (TMT) labeled peptides and high pH fractionation**. Cells were cultured as described above, $1 \times 10^6$ cells were collected by scraping from plates and pellets were washed twice with PBS. Sample preparation and labeling peptides with TMT isobaric mass tags was performed as per the manufacturer's instructions. Briefly, cells were lysed in 100 µL lysis buffer (10% SDS in 100 mM Triethyl ammonium bicarbonate (TEAB)) using strong ultra-sonication. Lysates were cleared by centrifugation at 16,000×g for 10 m at 4 °C and protein concentration was determined using the BCA protein assay kit (Thermo Scientific). Fifty micrograms of protein was reduced with 5 mM Tris 2-carboxyethylphosphine (TCEP) for 1 h 55 °C, and alkylated with 10 mM iodoacetamide for 30 min in the dark at 25 °C. Reduced and alkylated proteins were precipitated over night by adding six volumes of acetone at −20 °C. Acetone precipitated proteins were resuspended in 100 mM TEAB, pH 8.5 and digested by incubation with sequencing-grade modified trypsin overnight at 37 °C.

For TMT labeling, trypsinized peptide samples were subsequently labeled with isobaric tags (TMT 6-plex, Thermo Fisher Scientific). All samples were labeled with individual tags: RPE1 KO- TMT126, RM X- TMT127, RM 10;18- TMT128, RM 13- TMT129, RPE1 p53 KD- TMT130, RM 19p- TMT131.

After pooling the TMT labeled peptide samples, peptides were desalted on C18 reversed-phase columns and dried under vacuum. TMT-labeled peptides were fractionated by high-pH reversed-phase separation according to Wang et al.[58] using a YMC Triart C18 column (3 µm, 120 Å, 2.1 mm × 100 mm; YMC Co., Ltd., Japan) on an Agilent 1100 HPLC system. Peptides were loaded onto the column in buffer A (ammonium formate [10 mM, pH 10] in water) and eluted using a 30 min gradient from 3–90% buffer B (90% acetonitrile/10% ammonium formate [20 mM, pH 10]) at a flow rate of 0.3 mL/min. Elution of peptides was monitored with a UV detector (280 nm). For each replicate, 48 fractions were collected and pooled into four fractions using a post concatenation strategy. For each of the four replicates, 48 fractions were collected in a 96-well plate using high pH fractionation. The individual replicate fractions were pooled by combining fractions 1, 4, 8, 12 (…); 2, 5, 9, 13 (…); and so on into four resulting fractions, which were then dried under vacuum and subjected to LC-MS/MS[58].

The concatenated and TMT-labeled peptide mixtures were analyzed using nanoflow liquid chromatography (LC-MS/MS) on an EASY nano-LC 1200™ system (Thermo Fisher scientific), connected to a Q Exactive HF (Thermo Fisher scientific) through a Nanospray Flex Ion Source (Thermo Fisher Scientific). Three microliter of each fraction was separated on a 40 cm heated reversed phase HPLC column (75 µm inner diameter with a PicoTip Emitter™, New Objective) in-house packed with 1.9 µm C18 beads (ReproSil-Pur 120 C18-AQ, Dr. Maisch). Peptides were loaded in 5% buffer A (0.5% aequeous formic acid) and eluted with a 3 h gradient (5–95% buffer B (80% acetonitrile, 0.5% formic acid) at a constant flow rate of 0.25 µL/mL. Mass spectra were acquired in data dependent mode. Briefly, each full scan (mass range 375–1400 $m/z$, resolution of 60,000 at $m/z$ of 200, maximum injection time 80 ms, ion target of 3E6) was followed by high-energy collision dissociation based fragmentation (HCD) of the 15 most abundant isotope patterns with a charge state between 2 and 7 (normalized collision energy of 32, an isolation window of 0.7 $m/z$, resolution of 30,000 at $m/z$ of 200, maximum injection time 100 ms, AGC target value of 1E5, fixed first mass of 100 $m/z$ and dynamic exclusion set to 30 s).

MS data was processed with the MaxQuant software, version 1.6.3.3. All data was searched against the human reference proteome database (UniProt: UP000005640) with a peptide and protein FDR of less than 1%. All raw files as well as all MaxQuant output tables and parameters have been uploaded to PRIDE.

**Sample preparation for label-free quantified mass spectrometry**. The sample preparation for LFQ was performed similarly as for TMT labeling strategy, but following trypsin digestion, the digested peptides were desalted on C18 reversed-phase columns and dried under vacuum. Air dried peptides were analyzed using a nanoflow liquid chromatography (LC-MS/MS) on an EASY nano-LC 1200™ system (Thermo Fisher scientific), connected to a Q Exactive HF (Thermo Fisher scientific) through a Nanospray Flex Ion Source (Thermo Fisher Scientific). Two microliter of each fraction was separated on a 40 cm heated reversed phase HPLC column (75 µm inner diameter with a PicoTip Emitter™, New Objective) in-house packed with 1.9 µm C18 beads (ReproSil-Pur 120 C18-AQ, Dr. Maisch). Peptides were loaded in 5% buffer A (0.5% aequeous formic acid) and eluted with a 3 h gradient (5–95% buffer B (80% acetonitrile, 0.5% formic acid) at a constant flow rate of 0.25 µL/mL. Mass spectra were acquired in data dependent mode. Briefly, each full scan (mass range 300–1650 $m/z$, resolution of 60,000 at $m/z$ of 200,

maximum injection time 20 ms, ion target of 3E6) was followed by high-energy collision dissociation based fragmentation (HCD) of the 15 most abundant isotope patterns with a charge state between 2 and 7 (normalized collision energy of 28, an isolation window of 0.1.4 *m/z*, resolution of 15,000 at *m/z* of 200, maximum injection time 80 ms, AGC target value of 1.6E3, no fixed first mass and dynamic exclusion set to 20 s). MS data was processed with the MaxQuant software, version 1.6.3.3. All data was searched against the human reference proteome database (UniProt: UP000005640) with a peptide and protein FDR of less than 1%. All raw files as well as all MaxQuant output tables and parameters have been uploaded to PRIDE.

**Analysis of proteome data**. Identified protein groups were filtered to remove contaminants, reverse hits and proteins identified by site only. Next, Protein groups which were identified more than two times in at least one group of replicates (*N* = 4) were kept for further processing resulting in a set of 5887 Protein groups in total. For LFQ, Protein groups which were identified more than three times in at least one group of replicates (*N* = 4) were kept for further processing, resulting in a set of 5727 Protein groups in total Log2 TMT reporter intensities were cleaned for batch effects using the R package LIMMA[59] and further normalized using variance stabilization[60]. Next, protein intensities obtained from LFQ and TMT of mono-somic cell lines were normalized by shifting to diploid median and fold change calculation to the intensities of the wild type parental cell lines as described for the transcriptome analysis

**Combined analysis of genomic, transcriptomic, and proteomic datasets**. For further analysis comparing genomic, transcriptomic, and proteomic datasets, the DNA and mRNA datasets were matched to the corresponding protein entries and merged into a single table (Supplementary Data 1). To compare monosomic and trisomic cell lines, proteome data of trisomic RPE cell lines[25] was merged to the dataset. Chromosome/scaffold name, gene start (bp), gene stop (bp) and Ensembl gene stable ID (ENSG) were annotated through BioMart. Perseus was used to add additional annotation (GOBP, GOCC, CORUM) and to carry out 2D annotation enrichment analysis[61]. The figures showing Log2FC per chromosomes were generated using ggplot together with dplyr.

Density histograms were generated in R using the library k-density and EQL. The log2 ratios of the mRNA and proteome subsets were plotted as density histograms, including the median of both populations.

**TCGA and CCLE data analysis**. TCGA pan-cancer RNAseq data involving 11,060 samples were downloaded from Pan-Cancer Atlas and filtered to remove tumors that underwent whole genome doubling (WGD). Five thousand seven hundred and twenty-two primary tumor samples were kept for further analysis. RNAseq profiles of 418 samples from the Broad-Novartis Cancer Cell Line Encyclopedia (CCLE)[34], filtered to remove cell lines with WGD, were used for the analysis. Ploidy value annotated from[36] was used to define -somy status of pan-cancer dataset. Separation of ploidy values into Monosomy and Polysomy status relies was based on two thresholds. As there are no clear minima in the distribution of ploidy values clearly separating monosomic, disomic and polysomic samples, we assessed how the thresholds affect the final results by varying them from 1.66 to 1.90 (monosomy–disomy) and from 2.0 to 2.27 (disomy–polysomy). All thresholds resulted in a significant enrichment of p53 alterations in monosomic samples. CCLE sample ploidy value was retrieved from CCLE database. Ploidy values of both Pan-cancer and CCLE cohorts were inferred using ABSOLUTE algorithm[62] using the copy number data from https://data.broadinstitute.org/ccle/, file name:- CCLE_ABSOLUTE_combined_20181227.xlsx, sheet name: ABSOLUTE_combined.table.

Ribosome related pathway GO terms and KEGG pathway gene sets created by MsigDB6[63] were collected. ssGSEA[35] was applied on CCLE RNA-seq data to calculate pathway activity scores. A lower pathway score indicates that the genes in a specific pathway for a sample are under-expressed compared to the overall population.

TP53 classifier scores of TCGA pan-cancer cohorts were annotated from[36] as follows: TP53 score and ploidy of TCGA was collected from https://gdc.cancer.gov/about-data/publications/PanCan-DDR-2018, excel file name: TCGA_DDR_Data_Resources.xlsx, sheet name: DDR footprints.

The aneuploidy score was collected from https://www.sciencedirect.com/science/article/pii/S1535610818301119#app2, Table S2, file name: 1-s2.0-S1535610818301119-mmc2.xlsx.

The p53 score is based on a logistic regression model, where p53 functional inactivation status is the response and expression of genes are covariables. The logistic regression classifier is trained to estimate a set of parameters that can accurately predict the p53 alteration status. Given a new sample, this classifier uses a logistic sigmoid function to report a probability value representing to what degree p53 of a sample is likely to be altered, given the expression data of the sample.

**Statistical analysis**. Statistical analyses of cell biology data was performed in replicates and the number of the performed independent experiments are specified in respective figure legends. GraphPad Prism software was used for the statistic tests. Statistical analyses were performed using a two-tailed *T*-test or non-parametric *T*-

test as indicated in the corresponding figure legend. Values are shown as the mean ± sem or mean ± sd of multiple independent experiments. For imaging ana-lysis, the investigators were blinded to sample identity during assessment.

To identify significantly upregulated genes, a modified *T*-test adjusted for multiple testing (FDR = 0.05, S0 = 0.1, Perseus) was used to evaluate the normalized log2 mRNA intensities obtained for RM10ip53 cells treated with or without doxycycline.

Significance of TCGA and CCLE pathway score difference between different -somy groups was determined using one sided Wilcoxon rank sum test.

**Reporting summary**. Further information on research design is available in the Nature Research Reporting Summary linked to this article.

# Data availability

Source Data is provided with this paper. The TMT label mass spectrometry proteomics data have been deposited to the ProteomeXchange Consortium via the PRIDE partner repository with the dataset identifier PXD018440. LFQ data was deposited on PRIDE partner repository with the dataset identifier PXD022927. The transcriptome of RM X was uploaded on ENA with accession number. PRJEB38328 All other transcriptomes are uploaded in NCBI Omnibus with accession number. GSE150686 The human ribosome cryo-EM data used fort he ribosome structure presentation 6Y2L . The TCGA data used for the analysis are depositited at [https://api.gdc.cancer.gov/data/5dd5a767-8f9f-4579-abee-b1306a4d0ad2] The CCLE data use for the analysis are deposited at [https://data.broadinstitute.org/ccle/CCLE_ABSOLUTE_combined_20181227.xlsx] The Aneuploidy Scores used for the analysis are deposited at [https://www.sciencedirect.com/science/article/pii/S1535610818301119#app2] Source data are provided with this paper.

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

## Acknowledgements

The DNA and RNA sequencing was performed by the Genomics Core Facility EMBL, Heidelberg and by the NGS-Integrative Genomics Core Unit (NIG), Institute of Human Genetics, University Medical Center Göttingen (UMG). The knock down monosomic cell lines were generously provided by Rene Medema (National Cancer Institute, Amsterdam). We are thankful to the members of Storchova laboratory for their helpful discussions and critical comments on the work, and to Carina Heinrich, Philipp Elleringmann, Isabelle Kirchner, and Robin Roth for their help with the experimental work. We thank Jan Hauth from Fraunhofer Institute of Applied Mathematics in Kaiserslautern for his help with processing large datasets. This project was supported by DFG STO918-5/1 to Z.S. and FOR2800 to M.K., M.R., and Z.S.

## Author contributions

N.K.C. constructed and analyzed the monosomic ip53 cells lines, performed and evaluated the experiments, analyzed the bioinformatics data and wrote the manuscript; P.M. and M.R. performed the proteomics and analyzed the data; A.W. analyzed and validated the gene expression changes; B.R.M., C.B., and J.O.K. created the monosomic K.O. cell lines, performed the DNA sequencing and analysis; X.Z. and M.K. performed the CCLE and TCGA data mining; N.K.C., V.L.G., and F.W. performed polysome analyses; Z.S. conceived and supervised the study, analyzed the data and wrote the manuscript. All authors commented on the manuscript.

## Funding

## Competing interests

The authors declare no competing interests.
