## [Peer Review File · Nature Communications]

Reviewers' Comments:

Reviewer #1:

Remarks to the Author:

Chunduri and colleagues established an in vitro model to study the effects of monosomy in a p53-deficient cell lines. Functional studies revealed that monosomy both suppresses proliferation and drives genomic instability. Transcriptomic and proteomic profiling experiments showed that many genes affected by chromosome loss were compensated at the mRNA and/or protein level. These global profiling studies highlight the differences in genes/proteins regulated in monosomic vs trisomic cell lines. In particular, the authors observed a consistent decrease in ribosomal genes across multiple monosomic cell lines that was supported by both omics profiling data. This haploinsufficiency of ribosomal genes led to a decrease in translational activity and activated p53 downstream pathways when p53 was re-introduced ultimately inducing G1 arrest and therefore potentially explaining the decreased proliferation observed in monosomic cell lines. Finally, the authors leverage CCLE and TCGA datasets to support their findings of p53 loss and decreased ribosomal gene sets in monosomic cancer cell lines and in human tumors.

Although the findings are interesting, the data presented to support their claims are incomplete, lack proper quantification, and lack sufficient validation for omics profiling results. The authors also need to discuss the clinical implications of the study in more detail.

Major comments:

1. The authors need to better justify that the normalization method they used is appropriate for this study. This is important because a key conclusion of the study is that the gene dosage reduced due to monosomy is partially compensated by both transcriptional and posttranscriptional mechanisms. They need to make sure that the conclusion is not affected by the normalization method.
2. A common limitation of TMT is that the report ion ratios may be suppressed due to the coelution of TMT-labeled ions. Figures 3b and 3c showed more buffering at the protein level than at the RNA level. The authors need to make sure that this is not due to ratio suppression in TMT.
3. These omics findings should be validated by qPCR for selected mRNAs and western blotting/targeted MRM for selected proteins.
4. More buffering at the protein level than at the RNA level suggests a compensatory increase in protein translation, which seems to be contradictory to what reported later in the manuscript that the translation efficiency was low in these monosomic cells. This needs to be clarified.
5. The authors used RPL21 KD to recapitulate the effects of monosomy to p53. They need to show how RPL21 was reduced in their monosomic cell lines. How about ribosomal proteins encoded in other chromosomes (e.g. 10 and 19p), - can their depletion also fully account for the monosomic phenotype? Validating more than one ribosomal protein would help strengthen their conclusion that loss of ribosomal proteins bridges the monosomy to the biological outcome.
6. Their TCGA and CCLE validation suggested that "monosomy" cancer samples have a higher possibility of having a damaged p53 pathway. This is not unexpected, as previously pan-cancer study has shown that SCNA high tumors are enriched with TP53 mutation (PMID: 24071851). The way the authors defined "monosomy" for TCGA samples seemed to be arbitrary and might essentially pick up the samples with high SCNA tumors. Can they add more analysis to show that quantitative SCNA amount is equal in "monosomy" and "polysomy" samples but TP53 mutation is still higher in "monosomy" samples?
7. In the TCGA/CCLE analysis, it would also be interesting to separate monosomy samples involving ribosomal protein gene haploinsufficiency and the other monosomy samples for the TP53 mutation analysis.
8. One of the monosomic models characterized extensively throughout the paper is the monosomic RM10 cell line. Despite having an extra copy of 10q in all monosomic models as the authors point out in only a figure legend, the impact of this is not discussed elsewhere in the paper. Transcriptome and proteome plots are missing specifically for the RM10 cell line. An extra copy of 10q as shown in Fig 3b does not seem to impact mRNA and protein expression for all RM lines shown displayed in Figs 3a-b & S3a-f. Was this also the case for the RM10 line and could this extra copy of 10q account for the observed mRNA/protein compensation of genes/proteins located on this arm?
9. Although the authors make generous use of densitometry to quantify western blotting results,

details are unclear how the data were normalized (which bands from Ponceau were used for normalization) and there is a lack of quantification on selected data which is concerning. For example, the authors conclude "no uniform changes in cell cycle distribution in monosomic cell lines" and that "immunoblotting showed minor expression differences..." based on data in Fig S1b-c. However, quantification should be performed to fully support this conclusion since it does not readily appear to be differences by eye. There are also inconsistencies in cell cycle profiles between Figs S1b and S6d. In RM 10 -DOX cells in S6d, the percentage of G1 cells is higher than the experiment shown in for RM10 cells in S1b. This should be clarified since S6d claims an arrest in G1, where S1b claims no difference in cell cycle profiles compared to controls. The authors should also quantify the western blots in Fig S6a since the levels of the GAPDH loading varies dramatically throughout the blot and this was the basis for experiments performed throughout Fig 6.

10. For Rpl21 siRNA experiments in Figs 5g-h & 6f, the data is not compelling since the differences seen in quantification do not match the western blotting data. This precludes conclusions made about the effect of defective ribosome biogenesis mediating p53 activation. Higher quality blots/controls are needed to confirm Rpl21 is indeed the band detected by the antibody and that siRpl21 is not having off target effects.

11. Given the fact the authors mention how their studies may have identified ribosomal vulnerabilities induced by monosomy in cancer, there is lack of experiments to support the clinical utility of their findings. Also, since ribosomal genes are essential, discussion on ribosomal targeted therapeutic strategies should be included.

Minor comments:

1. In general, immunofluorescence images presented throughout the manuscript are not representative of the accompanied quantification. For example, in Fig 2a, control cells show complete lack of micronuclei and anaphase bridges despite the quantification in Fig2a-b depicting the presence of detectable baseline levels of both. Furthermore, the RM10 cell line chosen to display is quantified as being not significantly different from control cells in the % cells with micronuclei. The authors should show all images from all cell lines quantified and choose representative images that support their conclusions. Another example is the images chosen in Fig S6b. Showing only one cell in a field of view is not representative nor does it support a critical finding in the paper in S6c showing the inactivation of p53 over time induced by monosomy.
2. Although the authors only examined pathway enrichment supported by both mRNA and protein measurements, they should also consider unique pathways that are supported by one or the other since they may also lead to additional biological insights into mechanisms underlying monosomy.
3. Specific comments:
 - (a) Fig 1d: stats are missing to support significant growth defects induced by monosomic cell lines.
 - (b) Figs 1d-e & 2b-c: there seems to be inconsistencies in functional experiments between the RM13 KO and KD cell lines, this should be discussed in the text.
 - (c) Fig S1h-i: "HU" is not defined in the figure legend nor text and leads to inability to comprehend these figures
 - (d) Fig S2a: positive control cell line that is sensitive to 17-AAG is needed to confirm pharmacologic inhibition is working
 - (e) Fig S4d: Y-axis has two labels and should be corrected.

Reviewer #2:

Remarks to the Author:

In this study, the authors investigated the changes of gene expression in monosomies using multi-omics approaches and found a consistent reduction of ribosomal proteins (RPs) and impaired translation. In addition, the authors found that the haploinsufficiency of RP genes induces activation of the p53 pathway, suggesting a possible role of RPs in monosomic cells in cancer. I think the multi-omics data provided in this study are solid and valuable but not sufficient to claim the causative role of RPs in monosomic cancer. There are some inconsistencies that need to be validated or discussed.

1. In Fig. 4 and 5, the authors showed that RP gene expression and translation were downregulated in monosomic cells. However, the polysome analysis in Fig. 5e indicated that the amount of heavier polysomal fractions were elevated in the monosomic cells, which suggests the activation of global translation in the cells. This result is inconsistent with the notion of decreased translation in monosomies shown by the puromycin-incorporation assay. Since the activation of oncogenic pathways is known to induce overall protein synthesis and promotes cellular transformation, I think the upregulation of polysome formation could be possible consequence. The authors need to provide reasonable explanation on this and to discuss the discrepancy.

2. In Fig. 5g-j, the authors claimed that an RPL21 depletion reproduces the reduced translation occurred in chromosome 13 monosomy. To verify the results, rescue experiments such as introducing RPL21 mRNA into RM13 cell line will be required. Also, it is interesting to see the polysomal patterns to validate the translational activity in RPL21-depleted cells.

3. In Fig. 6, monosomic cells with restored p53 expression were analyzed. The authors concluded that incompatibility of functional p53 and monosomy is due to haploinsufficiency of RP genes. If so, what is happening in monosomy 7 and monosomy 21, where no RP gene is assigned and a loss of chr 7 is frequently observed in cancer.

Minor points

1. p.8, line 234: "In RM13, we observed increased level of SSU and reduced LSU".

I don't see any reduction of LSU in RM13 (Fig. 5e).

2. Fig. 6b: Why does the induced p53 level decreased so quickly in monosomic cells? The cells expressing ip53 may not recapitulate the physiological condition of p53 status in wild type.

3. Many studies have been already done in terms of ribosome defects and cancer etiology. I recommend to the authors to refer the following review papers: *Nature Reviews Cancer*, 19, 228-238, 2019; *Cold Spring Harb Perspect Biol*, 5, a012336, 2013; *WIREs RNA*, 2, 507-522, 2011.

Reviewer #3:

Remarks to the Author:

In this manuscript, Chunduri et al. generated several monosomic RPE-1 cell lines, and then utilize these cells to identify and investigate some attributes of monosomic cells. The authors first sought to compare and contrast these monosomies with a few previously generated trisomies, and found that although both display increased genomic instability and decreased cellular fitness, monosomic cell lines do not experience proteotoxic stress that is associated with some trisomic cell lines. They then surveyed both the transcriptome and proteome of the monosomic cells, and found that decreases in gene expression/protein levels, which would be expected from monosomy, are buffered at both the transcriptional and post-translational level. Analysis of the transcriptomic and proteomic data led to the finding that these monosomic lines often down-regulate the large/small ribosomal subunits. After looking further into ribosome biogenesis and translation, the authors suggest that decreased ribosome biogenesis necessitates loss of p53 for monosomies to proliferate. The study ends by providing some association analyses utilizing the CCLE and TCGA datasets.

This work represents an interesting attempt to study properties of monosomy in human cell lines. However, with only a few karyotypes in one immortalized cell line, it is far from comprehensive and the observations should not be generalized for monosomy. While some of the observations are intriguing, there is a lack of clear causal linkage or mechanistic insight. For example, ribosomal loss would be expected to globally decrease protein levels, yet the authors observe buffered protein levels higher than expected in their monosomies. Additionally, the role of p53 in the observed phenomena is unclear, as further discussed below. Finally, there are multiple important technical points that should be addressed to make the evidence much more compelling for the conclusions of this study.

Major Points:

1. The authors do not provide sufficient context and detail in the main text and methods on the generation of monosomic RPE-1 cell lines. This prevents a rigorous assessment of the validity and usefulness of the p53 knockout methodology in generating aneuploid cell lines, and hinders independent replication and future exploration by others in the field. p53 is a well-known factor that can limit the proliferation of aneuploid human cell lines (PMID: 20123995). Thus, it is not surprising that p53 deletion facilitates the survival and proliferation of certain karyotypes. p53 is also a key regulator of tetraploidy. The authors should provide a summary of karyotypes and aneuploidy/tetraploidy frequency obtained from their attempts to generate monosomic cell lines from both WT and p53^{-/-} RPE-1.

2. The authors should use trisomic cell lines generated with the same methodology as the monosomic cell lines in their experiments, instead of using trisomic lines generated by using microcell. The monosomic lines in this study are mostly p53^{-/-} while the trisomic lines are p53^{+/+}. p53 status could easily be a confounding factor in both the genomic (Figure 4) and biochemistry experiments (Figure 5) presented in the study.

3. How stable are these monosomic cell lines? Several studies have shown that aneuploid karyotypes can be quite unstable, which necessitates careful documentation of karyotypes at each stage of the study. The authors observed increased micronuclei formation in several lines, clearly indicating chromosomal instability. The stability and homogeneity of the monosomic cell lines are crucial for accurate interpretation of results. For instance, the partial dosage compensation observed could be simply explained by the authors inadvertently using a mixture of monosomic and euploid cells. As it stands, Supplementary Table 1 does not provide enough data to be convincing. The authors should provide data for at least 100 metaphase spreads per monosomic cell line, detailing both whether the expected monosomy is observed, as well as total chromosome counts.

4. Even if the karyotypes are stable and supports the observation of dosage compensation, the authors will need to provide a rational explanation that resolves the apparent conflict between the observed posttranscriptional dosage compensation and the observed ribosome biogenesis and protein translation defects. Otherwise these are just disjointed observations.

5. The authors argue that haploinsufficiency of ribosomal genes is responsible for the observed ribosomal biogenesis stress and p53 activation without providing sufficient experimental evidence. While siRNA-mediated RPL21 depletion phenocopied the ribosome biogenesis defect, the authors also should demonstrate that restoring RPL21 copy number in RM 13 ip53 is also sufficient to rescue the ribosome biogenesis defect. It will be helpful to provide a graphic detailing the chromosomal distribution of all human ribosomal protein genes. The conclusions could be further supported by including a second example, such as RPE-1 cells hemizygous for RPS24, which should phenocopy the ribosome biogenesis stress in RM 10 cells, presuming the authors' hypothesis holds. Figure 4e also suggests that trisomic lines could have impaired ribosome biogenesis, and so this phenotype may not be specific to monosomy.

6. Activation of the p53 pathway in these monosomic lines could be the result of DNA damage. The authors only looked at the impact of ribosome biogenesis on p53 activation, but Figure 2 shows clear signs of genomic instability, including γ H2AX accumulation and broken chromosomes trapped in micronuclei. This makes it unreasonable to conclude that the ribosomal biogenesis defect is the sole cause of p53 activation. The rescue experiment mentioned in point 4 above could help clarify.

7. While genome instability is observed in the monosomic cell lines, no mechanistic insight is provided. The observation by itself is not novel. Replication stress could possibly explain the observed DNA damage, anaphase bridges, and micronuclei, yet the authors did not see any difference in replication protein levels. The authors could examine this by directly measuring the progression of DNA replication forks with the DNA fiber assay.

8. The authors calculated a "p53 classifier score" and cited a previous publication (PMID: 29617664), but they did not provide any details on how they performed the calculation. The authors should also provide justification for their percentile cutoff when they assign the ploidy status of TCGA data in Figure 7. It is currently impossible to judge the validity of this analysis.

Reviewer #4:

Remarks to the Author:

I was asked to comment on the technical aspects of the proteome analysis and I will restrict my comments to this.

The choice of TMT labeling to profile quantitative changes in the proteomes of the investigated cell lines is fine but it comes with several caveats as the authors may be aware and which should be considered when interpreting the data.

It is well-known that the quantitative accuracy of TMT labeling suffers from an effect called 'ratio compression'. This means that the fold changes measured by this technique are almost always smaller than they truly are. As far as the current study is concerned, this is of particular relevance as

a) MS2 level quantification was employed on a QE-HF which cannot compensate the effect by an MS3 measurement

b) even though the proteome digests were separated in 48 hPH RP fractions, they were then combined into 4 prior to LC-MS/MS. This is unclear to this reviewer why one would want to do this. It will limit proteomic depth and almost entirely remove the opportunity to mitigate ratio compression.

In either case, the authors should explain/justify how they chose their fold-change criteria. It is difficult to guess if different cut-offs would change the interpretation of the data (or e. g. the GO analysis) and the authors should check this. Given that the authors performed replicate analysis for each of the cell lines, there is an opportunity to use both the magnitude of a change (fold-change) as well as the significance of that change (p-values) in the assessment of the data. The authors should perform such an analysis.

More information on the TMT labeling needs to be provided. At minimum, one would want to know which TMT label was used for which sample. It is unclear if replicates of the cell lines were multiplexed or if the cell lines were multiplexed etc. Perhaps, much of this is in the PRIDE submission but it should at least be possible to understand the experimental design of the proteome analysis.

Reviewers' comments:

Reviewer #1 (Remarks to the Author): expert in computational biology

Chunduri and colleagues established an in vitro model to study the effects of monosomy in a p53-deficient cell lines. Functional studies revealed that monosomy both suppresses proliferation and drives genomic instability. Transcriptomic and proteomic profiling experiments showed that many genes affected by chromosome loss were compensated at the mRNA and/or protein level. These global profiling studies highlight the differences in genes/proteins regulated in monosomic vs trisomic cell lines. In particular, the authors observed a consistent decrease in ribosomal genes across multiple monosomic cell lines that was supported by both omics profiling data. This haploinsufficiency of ribosomal genes led to a decrease in translational activity and activated p53 downstream pathways when p53 was re-introduced ultimately inducing G1 arrest and therefore potentially explaining the decreased proliferation observed in monosomic cell lines. Finally, the authors leverage CCLE and TCGA datasets to support their findings of p53 loss and decreased ribosomal gene sets in monosomic cancer cell lines and in human tumors.

Although the findings are interesting, the data presented to support their claims are incomplete, lack proper quantification, and lack sufficient validation for omics profiling results. The authors also need to discuss the clinical implications of the study in more detail.

Major comments:

1. The authors need to better justify that the normalization method they used is appropriate for this study. This is important because a key conclusion of the study is that the gene dosage reduced due to monosomy is partially compensated by both transcriptional and posttranscriptional mechanisms. They need to make sure that the conclusion is not affected by the normalization method.

>> Indeed, normalization strategies are key to any omics analysis, in particular in aneuploid cells, which we know from our long-standing experience with proteomics in aneuploid cells. Therefore, we have tried several normalization approaches, with strikingly similar results. We used technically normalized intensities transformed to the logarithmic scale after preprocessing in MQ. Both label free quantification that we now added (see PMID: 24942700), or corrected TMT reporter ion intensities (see PMID: 32892627) were normalized for batch effects by the R package limma (PMID: 25605792/). This is also as described in the methods. The data sets were normalized by shifting the individual replicates to the same diploid median to weight down the effect of the loss of chromosomes and the median was then shifted to zero. This step is a standard procedure performed to have ideal comparability between individual samples. Finally, we calculated fold changes to the diploid wild type cell line. Thus, throughout the manuscript, we always presented the log₂ fold changes (Log₂FC) relative to the parental control, as explained in the Material and Methods. These observed protein abundance changes were not affected by the normalization. In fact, the calculated ratios here are so robust that they outweigh the normalization steps before. For example, the 2D plots provide the same result even if we leave out the shifting to median, or when we calculate the FCs without normalization steps. The only noticeable difference is in the global heatmap of the TMT data before calculating the ratios when we leave out the batch effect removal, as then the reporter ions cluster together instead of the replicates. Moreover, as seen in the 2D plots (below), the results for TMT and LFQ are very similar, despite two different measurement and different technical normalization methods.

2. A common limitation of TMT is that the report ion ratios may be suppressed due to the coelution

of TMT-labeled ions. Figures 3b and 3c showed more buffering at the protein level than at the RNA level. The authors need to make sure that this is not due to ratio suppression in TMT.

>> Indeed, the problem with TMT labeling is well known. Therefore, we have performed from the beginning the proteome analysis with both the TMT labelling, as well as with label-free proteomics (LFQ). As one can see in a comparison of measurement of RM10;18 and RM13, both approaches yielded the same result (Figure R1). In the combined scatter plot for all monosomy encoded proteins/mRNA, both LFQ and TMT derived data show dosage compensation on protein level in comparison with the mRNA abundance (Fig. R1A). When considering dosage compensation in individual monosomic cell lines, you can see the example of RM10;18, where the median of the protein FCs related to monosomically encoded genes shifts only by 0.05 from TMT to LFQ (Fig. R1B). Finally, the 2D enrichments plots showed similar up- and downregulated pathways. In the first submission, we only used the TMT data because of the superior measurement coverage. With LFQ, the coverage was 71.48 %, while with TMT we obtained 88.65% coverage. We apologise for omitting the LFQ analysis in the previous manuscript version. The new data are now included in the manuscript, Fig. S4.

Figure R1. Comparison of proteomics data derived from TMT labelling (left) and LFQ (right). A. Scatter plot showing the \log_2 fold change (FC) in comparison to parental wild type of mRNA and proteins encoded on all monosomes (blue), and all disomes (yellow). The marginal density histograms show the distribution of

respective mRNAs and proteins. The expected median fold change of monosomic genes is marked by red dashed lines. The measured median fold changes of monosomic and disomic genes is marked by blue and yellow dashed lines, respectively. Note that the monosomy X was not evaluated by LFQ. B. Overlays of mRNA (dashed line) and protein (solid line) density histograms. Values of respective medians are plotted in the graph. Left: TMT, right: LFQ C. Two-dimensional annotation (2D) enrichment analysis based on the protein and mRNA changes in the monosomic cell line RM13 compared to RM 10;18. Each dot represents one category (GOBP, GOCC and chromosome location). Colors mark groups of related pathways as described in the inset. Axis-position represents scores of the pathways; negative values indicate downregulation, positive values indicate upregulation. Benjamini-Hochberg FDR Threshold < 0.02; left: TMT, right: LFQ.

3. These omics findings should be validated by qPCR for selected mRNAs and western blotting/targeted MRM for selected proteins.

>> While this request is completely understandable, we would like to point out that it is extremely difficult to validate by qPCR and immunoblotting abundance differences of 20-50 %, which is what we see for most of the monosomies and which is also often the variance of qPCR and western blotting. This is even stronger in aneuploid cells that are chromosomally unstable and have inherent non-genetic variability. Nevertheless, we have shown validation of several factors already in the previous submission. In figure S6a,b, we show validation of six different mRNAs by qPCR in several monosomic cell lines. Additionally, the ribosomal proteins RPL21 and RPS24 were also validated, as well as the differential regulation of the ribosomal and translational pathways.

4. More buffering at the protein level than at the RNA level suggests a compensatory increase in protein translation, which seems to be contradictory to what reported later in the manuscript that the translation efficiency was low in these monosomic cells. This needs to be clarified.

>> While we understand why the reviewer might interpret our data in this way, we believe that this is not possible. The buffering is observed only for the monosomic chromosome, while we observe a global reduction of translation. If this should contribute to buffering, however, it would mean that all proteins would be affected. Yet, the abundance of proteins encoded on disomic chromosomes is not altered when compared to parental diploid cells. The global decrease of translation rate, which affects all proteins, cannot be linked to the compensatory increase in the protein levels that are coded on only one specific chromosome. We actually never proposed this explanation in our manuscript and therefore, there is no contradiction.

Approximately 30-40% of proteins encoded on the monosomic chromosome are compensated at protein level. We believe that this is occurring either by selective increase in translation of specific proteins or by their reduced degradation (or both). In order to address this question, it would be essential to perform ribosome sequencing, or dynamic SILAC based proteomics. We have previously performed similar analysis in trisomic cells, with results that pointed out the importance of binding partners for stability of proteins (McShane et al, Cell 2016). The mechanisms involved in dosage compensation in monosomic cells are certainly very interesting, but this is clearly a new project, outside of the scope of this manuscript.

5. The authors used RPL21 KD to recapitulate the effects of monosomy to p53. They need to show how RPL21 was reduced in their monosomic cell lines. How about ribosomal proteins encoded in other chromosomes (e.g. 10 and 19p), - can their depletion also fully account for the monosomic

phenotype? Validating more than one ribosomal protein would help strengthen their conclusion that loss of ribosomal proteins bridges the monosomy to the biological outcome.

>> To address the reviewer’s request to analyze another ribosomal protein. We depleted, RPS24 , the only ribosomal protein encoded on chromosome 10. The depletion of RPS24 leads to reduced translation rate in diploid cells, comparably with cells with monosomy 10. Thus, the phenotype does not depend on the identity of the ribosomal gene. The new results are presented in Fig. S6c,d.

6. Their TCGA and CCLE validation suggested that “monosomy” cancer samples have a higher possibility of having a damaged p53 pathway. This is not unexpected, as previously pan-cancer study has shown that SCNA high tumors are enriched with TP53 mutation (PMID: 24071851). The way the authors defined “monosomy” for TCGA samples seemed to be arbitrary and might essentially pick up the samples with high SCNA tumors. Can they add more analysis to show that quantitative SCNA amount is equal in “monosomy” and “polysomy” samples but TP53 mutation is still higher in “monosomy” samples?

>> The reviewer raises an important question whether there is a potential confounding effect of aneuploidy, because the observed increase in p53 alterations in monosomy samples could be linked to the aneuploidy and not to the monosomy. Indeed, the monosomy samples tend to accumulate more chromosomal aberrations. The aneuploidy score fold change between monosomy samples and polysomy samples is 1.12 (Wilcoxon p=1.154e-10). To account for the aneuploidy effect and to check for an independent association between -somy status and p53 alteration, we used two different methods. First, we fitted a logistic regression model using the -somy status (monosomy as the base line) and the aneuploidy score as predictor variables and the p53 alteration status as binary response (altered vs wild-type). We also included an interaction term between -somy status and aneuploidy score. This clearly shows that the association of TP53 mutation and monosomy is not a confounding effect of high SCNA. These new results are now shown in figure 7c. The regression coefficients and confidence intervals can be seen below:

Coefficients		Estimate Std. Error	z value	Pr(> z)
Intercept	-0.228347	0.290728	-0.785	0.4322
AneuploidScore	0.040498	0.023374	1.733	0.0832
ploidy_typepolysomy	-1.781019	0.433336	-4.110	3.96e-05 ***
AneuploidyScore:ploidy_typepolysomy	-0.008841	0.034849	0.034849	0.7997

The two thresholds that are used for defining monosomic, disomic and polysomic samples are difficult to define. The thresholds we chose (monosomy: ploidy below 1.80, polysomy: ploidy above 2.19, WGD samples excluded) are based on the distribution of the ploidy in the samples. To test for the sensitivity of the results against the ploidy thresholds defining monosomy and polysomy, we varied the thresholds. Importantly, there was a significant logistic regression coefficient for ploidy-type when we varied these thresholds in the ranges from 1.66 to 1.90 (monosomy-disomy threshold) and from 2.0 to 2.27 (disomy-polysomy), respectively. This indicates that there is a significant

positive association between p53 alterations and monosomy, even after correcting for aneuploidy score effects and this association is robust against the thresholds used for defining the ploidy status.

In a second approach we defined an aneuploidy corrected p53 enrichment score (ACTES) as the ratio of p53 alteration frequency and aneuploidy score. We used bootstrap resampling to estimate a p-value for a positive difference in mean ACTES between monosomic and polysomic samples using 1000 bootstrap samples. The p-value is lower than $10e-3$. Again, this result is robust over a wide range of thresholds.

Overall, these results strongly support our statement that p53 alterations are significantly enriched in monosomy samples, accounting for an effect of aneuploidy. This result is robust against variations of ploidy-thresholds defining -somy groups.

7. In the TCGA/CCLC analysis, it would also be interesting to separate monosomy samples involving ribosomal protein gene haploinsufficiency and the other monosomy samples for the TP53 mutation analysis.

>> This is a very interesting idea and we thank the reviewer for this suggestion. We indeed filtered the monosomy category into two groups, with ribosomal levels lower than the median of the monosomy cohort (RPG low) and with higher levels (RPG high). This analysis revealed a significant difference of association with TP53 between these two groups ($p=7.9e-07$), as samples with RPG haploinsufficiency (RPG low) show more often TP53 mutation. This new analysis is shown in Fig. S8b.

8. One of the monosomic models characterized extensively throughout the paper is the monosomic RM10 cell line. Despite having an extra copy of 10q in all monosomic models as the authors point out in only a figure legend, the impact of this is not discussed elsewhere in the paper. Transcriptome and proteome plots are missing specifically for the RM10 cell line. An extra copy of 10q as shown in Fig 3b does not seem to impact mRNA and protein expression for all RM lines shown displayed in Figs 3a-b & S3a-f. Was this also the case for the RM10 line and could this extra copy of 10q account for the observed mRNA/protein compensation of genes/proteins located on this arm?

>> This remark is due to a misunderstanding of the consequence of the used normalization, which we now attempt to explain better. Importantly, the gain of 10q is an SCNA typical for RPE1 cell line and therefore it is present in all cell lines, as seen in Fig. 1b. Since the transcriptomics and proteomics plots are presented as fold changes relative to the parental cell line, which also carries the extra 10q, there will be no apparent impact on mRNA and protein expression when plots normalized to parental genome are shown. This is explained in the text, in the figure legends and also in the Material and Methods. Therefore, the extra copy of 10q cannot account for the mRNA/protein compensation. This is further supported by the fact that the observed mRNA/protein compensation does not affect only genes located on 10q arm. We now attempted to improve our explanation of the normalization strategy.

9. Although the authors make generous use of densitometry to quantify western blotting results, details are unclear how the data were normalized (which bands from Ponceau were used for normalization) and there is a lack of quantification on selected data which is concerning. For example, the authors conclude “no uniform changes in cell cycle distribution in monosomic cell lines” and that “immunoblotting showed minor expression differences...” based on data in Fig S1b-c. However, quantification and should be performed to fully support this conclusion since it does readily appear to be differences by eye. There is also inconsistencies in cell cycle profiles between Figs S1b and S6d. In RM 10 -DOX cells in S6d, the percentage of G1 cells is higher than the experiment

shown in for RM10 cells in S1b. This should be clarified since S6d claims an arrest in G1, where S1b claims no difference in cell cycle profiles compared to controls. The authors should also quantify the western blots in Fig S6a since the levels of the GAPDH loading varies dramatically throughout the blot and this was the basis for experiments performed throughout Fig 6.

>> We apologize for the lacking quantification. The reason why we did not provide these quantifications is that there were no “uniform” differences, meaning no differences that would be shared by all monosomies. We now actually remove the plot, as it is not relevant to our finding and does not significantly contribute to the key message of our manuscript. The difference between the cell cycle profiles in RM 10 -DOX might be caused by two problems. First, this is a new cell line that was generated by transfection and a single-cell clone was selected for antibiotic resistance during the process for generating doxycycline inducible cell line. Due to this lengthy process that takes several generation, additional chromosomal changes may have occurred due to the increased genomic instability of monosomic cells. Second, there might be a low expression of p53 due to a leaky inducible promoter. However, despite these differences between the cell lines used in S1b and S7e (previously S6d, note that the cell lines are congenic, but not identical, as explained above), our description and interpretation of the results is correct: there is a G1 arrest in monosomic cells with functional p53.

For the normalization of western blotting results, we used Ponceau staining as indicated in the respective figure legends. Since selection of a single band from the Ponceau staining could possibly bias the quantification, we generally use a large region of the entire membrane that contains several bands between 35 to 60 kD for normalization. We will add this explanation to Material and Methods.

10. For Rpl21 siRNA experiments in Figs 5g-h & 6f, the data is not compelling since the differences seen in quantification do not match the western blotting data. This precludes conclusions made about the effect of defective ribosome biogenesis mediating p53 activation. Higher quality blots/controls are needed to confirm Rpl21 is indeed the band detected by the antibody and that siRpl21 is not having off target effects.

>> We apologize for this discrepancy. We selected a new representative of western blotting and provided higher quality blots. The RPL21 antibody used for western blotting was verified using the cell lysates from RM 13 (Monosomy 13, RPL21 gene is encoded on chr.13), which should express lower levels of Rpl21 protein, as seen in Figure 5g.

11. Given the fact the authors mention how their studies may have identified ribosomal vulnerabilities induced by monosomy in cancer, there is lack of experiments to support the clinical utility of their findings. Also, since ribosomal genes are essential, discussion on ribosomal targeted therapeutic strategies should be included.

>> This is certainly an interesting aspect of our study, but in fact this is not a focus of the presented research, where we rather attempted a first systematic analysis of consequences of monosomy in human cells. We are confident that experiments to support the clinical utility of these findings require a new project and are beyond the scope of this manuscript. In fact, we have already initiated further work in this direction. We successfully established a cell competition assay for identifying differential sensitivities of monosomies to various drugs and also identified first drug that differentially affects monosomic cell lines. This manuscript (<https://www.biorxiv.org/content/10.1101/2020.09.25.314229v1.full>) is currently reviewed elsewhere. Finally, we want to emphasize that the role of monosomy in cancer is not the main topic

of our manuscript. Rather, we address the general cellular consequences of monosomy. In the newly submitted version, we changed the title accordingly to clearly point out the focus.

Minor comments:

1. In general, immunofluorescence images presented throughout the manuscript are not representative of the accompanied quantification. For example, in Fig 2a, control cells show complete lack of micronuclei and anaphase bridges despite the quantification in Fig2a-b depicting the presence of detectable baseline levels of both. Furthermore, the RM10 cell line chosen to display is quantified as being not significantly different from control cells in the % cells with micronuclei. The authors should show all images from all cell lines quantified and choose representative images that support their conclusions. Another example is the images chosen in Fig S6b. Showing only cell in a field of view is not representative nor does it support a critical finding in the paper in S6c showing the inactivation of p53 over time induced by monosomy.

>> We apologize for the image selection. We now added new and additional images to support our claims, such as an entire field with several cells for the micronuclei formation etc. These new images are part of figure 2a, S1d and S7c.

2. Although the authors only examined pathway enrichment supported by both mRNA and protein measurements, they should also consider unique pathways that are supported by one or the other since they may also lead to additional biological insights into mechanisms underlying monosomy.

>> We are not sure how pathway that would be deregulated only on mRNA level, but not on protein level, would provide a new insight. In the first submission, we presented plots comparing deregulated pathways on transcriptome and on proteome level (S5b,c). Indeed, there are some pathways differentially regulated on these levels, yet, they are not shared among multiple monosomies. Since we focused on common features of monosomies, we felt that this is beyond the scope of the manuscript and the text would become too unfocused.

3. Specific comments:

(a) Fig 1d: stats are missing to support significant growth defects induced by monosomic cell lines.

>> We added the statistical analysis.

(b) Figs 1d-e & 2b-c: there seems to be inconsistencies in functional experiments between the RM13 KO and KD cell lines, this should be discussed in the text.

>> We noted in the discussion the differences.

(c) Fig S1h-i: "HU" is not defined in the figure legend nor text and leads to inability to comprehend these figures

>> We added the definitions of this abbreviation.

(d) Fig S2a: positive control cell line that is sensitive to 17-AAG is needed to confirm pharmacologic inhibition is working

>> As the cells are dying in presence of 17-AAG, we are confident that the pharmacologic inhibition is working.

(e) Fig S4d: Y-axis has two labels and should be corrected.

>> The labeling was corrected.

Reviewer #2 (Remarks to the Author): expert in ribosomal biogenesis

In this study, the authors investigated the changes of gene expression in monosomies using multi-omics approaches and found a consistent reduction of ribosomal proteins (RPs) and impaired translation. In addition, the authors found that the haploinsufficiency of RP genes induces activation of the p53 pathway, suggesting a possible role of RPs in monosomic cells in cancer. I think the multi-omics data provided in this study are solid and valuable but not sufficient to claim the causative role of RPs in monosomic cancer. There are some inconsistencies that need to be validated or discussed.

1. In Fig. 4 and 5, the authors showed that RP gene expression and translation were downregulated in monosomic cells. However, the polysome analysis in Fig. 5e indicated that the amount of heavier polysomal fractions were elevated in the monosomic cells, which suggests the activation of global translation in the cells. This result is inconsistent with the notion of decreased translation in monosomies shown by the puromycin-incorporation assay. Since the activation of oncogenic pathways is known to induce overall protein synthesis and promotes cellular transformation, I think the upregulation of polysome formation could be possible consequence. The authors need to provide reasonable explanation on this and to discuss the discrepancy.

>> This is indeed an interesting point. We explain this by the fact that the monosomic cells decrease the translation rate, but not the total protein levels, as can be recognized by the fact that not all proteins are downregulated compared to the isogenic diploid, but only a small fraction of specific categories. Thus, one conclusion is that they require longer time to synthesize the proteins, but the final amount is not affected. This matches with our finding that the proliferation of monosomic cells is delayed. The accumulation of heavier polysomic fraction may be counter-intuitive, but could actually have a very simple reason. Due to the reduced amounts of L21 or S24, one can expect that a number of ribosomal subunits lack these proteins. Some of these incomplete subunits may still engage in translation. The incomplete subunits may have problems translating a transcript, which might be expressed by reduced elongation speed. If this is the case, only one slow ribosome on a transcript would actually lead to a general slowdown of translation of this specific transcript, since ribosomes initiating the same transcript after the 'slow' ribosome cannot outrun their preceding ribosome. So, even only a very small proportion of defective ribosomes could slow down elongation of the complete ribosome pool. And this finally, would be observed as a larger amount of polysomes compared with the wild type. In future we hope to be able to identify the involved mechanism. We now added following text to address the polysome discrepancy:

“Interestingly, the monosomic cell lines appear to contain more heavy polysomes than the parental control. While the increased polysome peak suggests increased ribosome numbers on mRNA, whether this is due to compensatory increase in the translation or whether it is caused by ribosome stalling leading to accumulation of multiple ribosomes on mRNA remains to be evaluated in future.”

2. In Fig. 5g-j, the authors claimed that an RPL21 depletion reproduces the reduced translation

occurred in chromosome 13 monosomy. To verify the results, rescue experiments such as introducing RPL21 mRNA into RM13 cell line will be required. Also, it is interesting to see the polysomal patterns to validate the translational activity in RPL21-depleted cells.

>> We agree with the reviewer that this would be a useful addition to the manuscript. We now added data from analysis of second ribosomal gene (RPS24), where its depletion causes the same phenotype as seen in monosomy of the respective chromosome (chr.10). However, we were not able to restore the expression of ribosomal genes, despite several months of efforts. We are convinced that this experiment is not possible with currently available technology. In fact, recently published papers show that ribosomal genes cannot be overexpressed, most likely due to a tight control of the stoichiometry of ribosomal proteins (e.g. PMID: 27552055, PMID: 31142641, PMID: 27385339). In line with these findings, we observed that overexpressed ribosomal factors are quickly degraded by proteasome, and accumulate in an insoluble fraction when proteasome is inhibited. As we cannot directly confirm the link between haploinsufficiency and monosomy, we have renamed our manuscript and toned down this specific conclusion in the text. Additionally, we added following text to our Discussion:

“An ultimate test for our hypothesis would be a rescue of the translation defect in monosomic cell by restoring the levels of ribosomal proteins. Our attempts to rescue the RPL21 expression were not successful, as the excessive protein was readily degraded by proteasome (NKC, personal communication). This is likely due to tight regulation of RPG expression that renders individual ribosome subunits generally resistant to overexpression. Novel approaches will have to be developed to perform this experiment in future.”

While we understand the request for polysome profiling of RPL2-depleted wild type cells, we were not able to perform these experiments, as scaling up the siRNA experiments for the amounts of cells required for polysome profiling is beyond feasibility.

3. In Fig. 6, monosomic cells with restored p53 expression were analyzed. The authors concluded that incompatibility of functional p53 and monosomy is due to haploinsufficiency of RP genes. If so, what is happening in monosomy 7 and monosomy 21, where no RP gene is assigned and a loss of chr 7 is frequently observed in cancer.

>> We do not see a contradiction here, since we do not claim that monosomy is a causative trigger of cancer. Our conclusion is that cancers that have monosomies are enriched for p53 inactivating mutations because of the haploinsufficiency of RP genes. In fact, one could argue that the frequent occurrence of monosomy 7 in hematopoietic cancers (monosomy 7 is rare in solid tumors) is due to the lack of the RP genes on chr.7, since they do not suffer from the haploinsufficiency of RP genes. Additionally, we never argue that the RP gene haploinsufficiency is the only important aspect. In case of monosomy 7, haploinsufficiency of other genes may contribute positively to tumorigenesis specifically in hematopoietic cancers. This is also supported by the fact that monosomy 21 is rather rare in cancer. Thus, the presence/absence of RPG possibly does not affect the role of specific monosomy in cancer. Importantly, we want to emphasize that a causative role of monosomy in cancer is not a topic of our manuscript. Rather, we address the general cellular consequences of monosomy.

Minor points

1. p.8, line 234: “ In RM13, we observed increased level of SSU and reduced LSU”.
I don't see any reduction of LSU in RM13 (Fig. 5e).

>> We apologize for this statement; indeed, in the presented plot the LSU is reduced in comparison to RM10, but remains the same in comparison with the parental cell lines. However, the ratio of LSU to SSU is consistently altered in monosomies. We amended the text as follows :

“The polysome profiling revealed changes in the ratio of unassembled large and small ribosomal subunits when compared to diploid control (Fig. 5e, f). We observed a reduction of SSU and accumulation of LSU in RM 10 when compared to diploid control, which is consistent with RPS24 (small ribosomal subunit protein 24) being the only RPG encoded on Chr.10 (Fig. 5e, f). In RM13 that encodes only LSU protein RPL21, we observed increased levels of SSU, while LSU did not change.”

2. Fig. 6b: Why does the induced p53 level decreased so quickly in monosomic cells? The cells expressing ip53 may not recapitulate the physiological condition of p53 status in wild type.

>> It is indeed conceivable that the ip53 does not perfectly recapitulate the physiological conditions of wild type p53, however, this is the only way how to perform the experiment since the expression of p53 is toxic in monosomic cells. There are two important aspects that suggest that the ip53 does not affect the results. First, the dynamics of p53 expression in the parental wild type cells is not altered, the expression does not decrease and the cells normally proliferate. Second, the protein p53 is highly unstable and degrades very quickly and we observed that the degradation rates were similar in ip53 and in p53.

3. Many studies have been already done in terms of ribosome defects and cancer etiology. I recommend to the authors to refer the following review papers: Nature Reviews Cancer, 19, 228-238, 2019; Cold Spring Harb Perspect Biol, 5, a012336, 2013; WIREs RNA, 2, 507-522, 2011.

>> We thank the reviewer for these valuable suggestions, but we do not wish to expand the discussion in this direction. While this topic is indeed extremely interesting, we rather wish to keep a strong focus of the manuscript on the consequences of monosomy. We strongly feel that further discussion should be reserved to future projects, where more relevant results will be obtained from monosomic model cell lines and cancers.

Reviewer #3 (Remarks to the Author): expert in aneuploidies

In this manuscript, Chunduri et al. generated several monosomic RPE-1 cell lines, and then utilize these cells to identify and investigate some attributes of monosomic cells. The authors first sought to compare and contrast these monosomies with a few previously generated trisomies, and found that although both display increased genomic instability and decreased cellular fitness, monosomic cell lines do not experience proteotoxic stress that is associated with some trisomic cell lines. They then surveyed both the transcriptome and proteome of the monosomic cells, and found that decreases in gene expression/protein levels, which would be expected from monosomy, are buffered at both the transcriptional and post-translational level. Analysis of the transcriptomic and proteomic data led to the finding that these monosomic lines often down-regulate the large/small ribosomal subunits. After looking further into ribosome biogenesis and translation, the authors suggest that decreased ribosome biogenesis necessitates loss of p53 for monosomies to proliferate. The study ends by providing some association analyses utilizing the CCLE and TCGA datasets.

This work represents an interesting attempt to study properties of monosomy in human cell lines.

However, with only a few karyotypes in one immortalized cell line, it is far from comprehensive and the observations should not be generalized for monosomy. While some of the observations are intriguing, there is a lack of clear causal linkage or mechanistic insight. For example, ribosomal loss would be expected to globally decrease protein levels, yet the authors observe buffered protein levels higher than expected in their monosomies. Additionally, the role of p53 in the observed phenomena is unclear, as further discussed below. Finally, there are multiple important technical points that should be addressed to make the evidence much more compelling for the conclusions of this study.

Major Points:

1. The authors do not provide sufficient context and detail in the main text and methods on the generation of monosomic RPE-1 cell lines. This prevents a rigorous assessment of the validity and usefulness of the p53 knockout methodology in generating aneuploid cell lines, and hinders independent replication and future exploration by others in the field. p53 is a well-known factor that can limit the proliferation of aneuploid human cell lines (PMID: 20123995). Thus, it is not surprising that p53 deletion facilitates the survival and proliferation of certain karyotypes. p53 is also a key regulator of tetraploidy. The authors should provide a summary of karyotypes and aneuploidy/tetraploidy frequency obtained from their attempts to generate monosomic cell lines from both WT and p53^{-/-} RPE-1.

>> This is a misunderstanding and we apologize that our description in the manuscript allowed this interpretation. We do not offer this as a novel methodology specific for generation of monosomic cells. Rather, indeed, the monosomy arises from chromosome missegregation and its survival is permitted thanks to the loss of p53. We never claimed that the monosomy arose as a direct consequence of targeting p53. We also do not call it “surprising”. We just simple state that after deletion of p53, some of the clones were monosomic (and many have other aberrations); we used the monosomic clones for our analysis. Single cell sequencing performed in the cells immediately following missegregation and published by Soto et al, 2017(Medema laboratory, PMID: 28636931, cited in our manuscript, note that the KD clones were a kind gift from Rene Medema) shows the frequency of various karyotypic changes. This information does not affect the conclusions of our manuscript. We now rewrote the part describing how we obtained the clones to avoid this misinterpretation.

2. The authors should use trisomic cell lines generated with the same methodology as the monosomic cell lines in their experiments, instead of using trisomic lines generated by using microcell. The monosomic lines in this study are mostly p53^{-/-} while the trisomic lines are p53^{+/+}. p53 status could easily be a confounding factor in both the genomic (Figure 4) and biochemistry experiments (Figure 5) presented in the study.

>> We agree with the reviewer and thank for pointing this out. Unfortunately, it has not been possible for us to obtain isogenic RPE1 cell with a single trisomy. As can be seen in the publication by Soto et al, 2017, p53^{-/-} clones with trisomies always accumulated additional aberrations. This is likely because the loss of p53 allows accumulation of multiple karyotypic changes. In the new version, we decided to remove this data, since, as pointed by the reviewer, the comparison is imperfect. Instead, we focus solely on the monosomy. We only point out in the discussion that the response to monosomy appears to differ from response to trisomy.

We would like to stress out that the presence or absence of p53 is not a strong confounding factor in the cellular response at least to monosomy, as can be seen in the figure 6d, where hierarchical

clustering of the transcriptome reveals that specific monosomies cluster together regardless their p53 status.

3. How stable are these monosomic cell lines? Several studies have shown that aneuploid karyotypes can be quite unstable, which necessitates careful documentation of karyotypes at each stage of the study. The authors observed increased micronuclei formation in several lines, clearly indicating chromosomal instability. The stability and homogeneity of the monosomic cell lines are crucial for accurate interpretation of results. For instance, the partial dosage compensation observed could be simply explained by the authors inadvertently using a mixture of monosomic and euploid cells. As it stands, Supplementary Table 1 does not provide enough data to be convincing. The authors should provide data for at least 100 metaphase spreads per monosomic cell line, detailing both whether the expected monosomy is observed, as well as total chromosome counts.

>> As we state in our manuscript, the monosomic cell lines are less stable than the parental diploids and indeed, heterogeneity and subtle changes in karyotype documented in our manuscript can be observed. Thanks to our long standing experience with aneuploid cells, we have developed an approach to handle these cells. We always expand the freshly created aneuploid cells and freeze as many batches as possible. These cells are sequenced and, if the karyotype was confirmed, then only these batches are used for experiments, and for every experiment a new batch is taken. We do not use samples with mosaic karyotypes and therefore the heterogeneity of the monosome is not more than ~10 %. (standard NGS calling threshold) We never passage the cells for more than 3-4 passages. Additionally, we disagree that the dosage compensation could be simple explained by a mixture of monosomic and euploid cells. In such case one would expect the mRNAs and proteins to be changed identically (or at least similarly), which is not the case, as the dosage compensation observed here occurs dominantly on protein level.

4. Even if the karyotypes are stable and supports the observation of dosage compensation, the authors will need to provide a rational explanation that resolves the apparent conflict between the observed posttranscriptional dosage compensation and the observed ribosome biogenesis and protein translation defects. Otherwise these are just disjointed observations.

>> In fact, we are confident that these are disjointed observations; two independent outcomes of monosomy. In our manuscript, we never linked these two observations together. On one hand, we show that there is posttranscriptional dosage compensation that occurs by so far unidentified mechanism that may, but does not have to be linked directly to translation. Actually, we rather hypothesize that altered protein stability might be involved in the dosage compensation. Addressing this aspect is important and we plan it in the future, however, it will require a marked effort and specialized technologies, similar as we used in our paper addressing the dosage compensation in trisomic cells (McShane et al, Cell 2016, PMID: 27720452). Second independent finding is that monosomic cells suffer from translation defects, possibly caused by haploinsufficiency of ribosomal proteins. The translational defects are not related and responsible for the compensation – we never proposed it in the manuscript and we do not consider it a valid hypothesis. We also do not see any discrepancy here, since the dosage compensation concerns only genes localized on the individual monosome, while the reduced translation due to low ribosome number would affect the global proteome.

5. The authors argue that haploinsufficiency of ribosomal genes is responsible for the observed ribosomal biogenesis stress and p53 activation without providing sufficient experimental evidence. While siRNA-mediated RPL21 depletion phenocopied the ribosome biogenesis defect, the authors

also should demonstrate that restoring RPL21 copy number in RM 13 ip53 is also sufficient to rescue the ribosome biogenesis defect. It will be helpful to provide a graphic detailing the chromosomal distribution of all human ribosomal protein genes. The conclusions could be further supported by including a second example, such as RPE-1 cells hemizygous for RPS24, which should phenocopy the ribosome biogenesis stress in RM 10 cells, presuming the authors' hypothesis holds. Figure 4e also suggests that trisomic lines could have impaired ribosome biogenesis, and so this phenotype may not be specific to monosomy.

>> We agree with the reviewer that this would be a useful addition to the manuscript. As requested, we now added data from analysis of second ribosomal gene (RPS24), where its depletion causes the same phenotype as seen in monosomy of the respective chromosome (chr.10). However, we were not able to restore the expression of ribosomal genes, despite several months of efforts. We are convinced that this experiment is not possible with currently available technology. In fact, recently published papers show that ribosomal genes cannot be overexpressed, most likely due to a tight control of the stoichiometry of ribosomal proteins (e.g. PMID: 27552055, PMID: 31142641, PMID: 27385339). In line with these findings, we observed that overexpressed ribosomal factors are quickly degraded by proteasome, and accumulate in an insoluble fraction when proteasome is inhibited. As we cannot directly confirm the link between haploinsufficiency and monosomy, we renamed our manuscript and toned down this specific conclusion in the text.

6. Activation of the p53 pathway in these monosomic lines could be the result of DNA damage. The authors only looked at the impact of ribosome biogenesis on p53 activation, but Figure 2 shows clear signs of genomic instability, including γ H2AX accumulation and broken chromosomes trapped in micronuclei. This makes it unreasonable to conclude that the ribosomal biogenesis defect is the sole cause of p53 activation. The rescue experiment mentioned in point 4 above could help clarify.

>> We agree that this is a possible interpretation, although not very plausible. There are two reasons for our conclusions: 1. We did not observe an activation of replication and DNA damage checkpoints, suggesting that the levels of DNA damage are rather low, not sufficient to trigger an arrest. 2. We did not observe any increase in γ H2AX accumulation in monosomy of chromosome X (RMX, figure 2E), nor accumulation of micronuclei etc. Yet, the proliferation is reduced and the translation, too. Of note, RPS4X, a RP gene encoded on chromosome X that is not subject to X chromosome inactivation and its haploinsufficiency has been recognized to possibly contribute to the Turner syndrome phenotype (45, X). We now, however, added this possible explanation (which is not mutually exclusive with the role of ribosomal insufficiency) to our Discussion.

7. While genome instability is observed in the monosomic cell lines, no mechanistic insight is provided. The observation by itself is not novel. Replication stress could possibly explain the observed DNA damage, anaphase bridges, and micronuclei, yet the authors did not see any difference in replication protein levels. The authors could examine this by directly measuring the progression of DNA replication forks with the DNA fiber assay.

>> The observation that monosomic human cell lines are genomically unstable has not been, to my knowledge, previously published. I would be thankful if the reviewer could provide the respective citation. To our knowledge, this is the first systematic study of monosomy in human cells. Indeed, replication stress that we and others showed to cause genomic instability in trisomic cells would be the first candidate, but the results have been so far negative. Further analysis is way beyond the focus of the manuscript and would require much more than a simple DNA fiber assay, which per se is anyway not sufficient to provide mechanistic insight to causes of genome instability.

8. The authors calculated a “p53 classifier score” and cited a previous publication (PMID: 29617664), but they did not provide any details on how they performed the calculation. The authors should also provide justification for their percentile cutoff when they assign the ploidy status of TCGA data in Figure 7. It is currently impossible to judge the validity of this analysis.

>> We apologize for providing insufficient information about the details, as this is described in a previously published work. We now added the details as well as further explanation for the used cutoff to enable the readers to judge the analysis. Regarding the cutoff thresholds, we performed a sensitivity analysis to assess how sensitive the results are to the thresholds chosen. The two thresholds that are used for defining monosomic, disomic and polysomic samples are difficult to define. The thresholds we chose (monosomy: ploidy below 1.80, polysomy: ploidy above 2.19, WGD samples excluded) are based on the distribution of the ploidy in the samples. To test for the sensitivity of the results against the ploidy thresholds defining monosomy and polysomy, we varied the thresholds. Importantly, there was a significant logistic regression coefficient for ploidy-type when we varied these thresholds in the ranges from 1.66 to 1.90 (monosomy-disomy threshold) and from 2.0 to 2.27 (disomy-polysomy), respectively. This indicates that there is a significant positive association between p53 alterations and monosomy, even after correcting for aneuploidy score effects and this association is robust against the thresholds used for defining the ploidy status.

In a second approach we defined an aneuploidy corrected p53 enrichment score (ACTES) as the ratio of p53 alteration frequency and aneuploidy score. We used bootstrap resampling to estimate a p-value for a positive difference in mean ACTES between monosomic and polysomic samples using 1000 bootstrap samples. The p-value is lower than $10e^{-3}$. Again, this result is robust over a wide range of thresholds. Thus, the results are independent of the thresholds.

Reviewer #4 (Remarks to the Author): expert in proteomics

I was asked to comment on the technical aspects of the proteome analysis and I will restrict my comments to this.

The choice of TMT labeling to profile quantitative changes in the proteomes of the investigated cell lines is fine but it comes with several caveats as the authors may be aware and which should be considered when interpreting the data.

It is well-known that the quantitative accuracy of TMT labeling suffers from an effect called 'ratio compression'. This means that the fold changes measured by this technique are almost always smaller than they truly are. As far as the current study is concerned, this is of particular relevance as

- MS2 level quantification was employed on a QE-HF which cannot compensate the effect by an MS3 measurement
- even though the proteome digests were separated in 48 hPH RP fractions, they were then combined into 4 prior to LC-MS/MS. This is unclear to this reviewer why one would want to do this. It will limit proteomic depth and almost entirely remove the opportunity to mitigate ratio compression.

>>We are aware that quantification with TMT mass labels suffers from ratio compression. Therefore, we compared proteome changes by TMT labelling or a label-free quantification (LFQ) strategies. As

the results are very comparable (see the response to reviewer 1), the impact of ratio compression appears to be at most very modest. We believe that this is due to the high sequence coverage that can be obtained with the **fast acquisition speed, high resolution** as well as the **efficient precursor selection** capabilities provided by the advanced quadrupole technology of the Q Exactive HF instrument series. Although we combined the fractionated peptides into only four pools, we used 3 hour gradients to analyze each pool of peptides. Because of the extensive catenation strategy, the peptides in all pools eluted very evenly across the entire gradient, thereby providing an effective strategy to minimize co-elution of precursors. We believe this strategy to be as effective in reducing co-elution as analyzing a higher number of fractions with shorter gradients. There are numerous publications in which TMT labeling experiments are carried out on Q Exactive HF instruments.

In either case, the authors should explain/justify how they chose their fold-change criteria. It is difficult to guess if different cut-offs would change the interpretation of the data (or e. g. the GO analysis) and the authors should check this. Given that the authors performed replicate analysis for each of the cell lines, there is an opportunity to use both the magnitude of a change (fold-change) as well as the significance of that change (p-values) in the assessment of the data. The authors should perform such an analysis.

>> As mentioned above (response to reviewer 1), we have performed also the label free analysis (LFQ) that provided similar data as the TMT analysis, but with a much better coverage. Therefore, we used the TMT data for further analysis. However, we will add the results from LFQ analysis to the manuscript. Concerning the GO analysis and the different cut offs, and p values, we in fact use both the fold-change and the p-values for the analysis. This is explained in the Material and Methods. For the 2D enrichment of GO terms (Fig. 4), the cutoff was set to 0.02 Benjamini-Hochberg corrected false discovery rate (FDR). The individual FDRs can be found in the Supplementary table 4 together with the p-values. For example, the p-value of GO term "large ribosomal subunit" in the 2D enrichment of RM10;18 vs RM13 is at $-3.81E-07$ and the BH-FDR at $2.09E-05$. Thus, so different cutoffs within a reasonable scope would not change our results. We use a cutoff of 0.02, which rather strict, and complement the assessment of the data with additional analysis of the p-values.

Figure R1. Comparison of proteomics data derived from TMT labelling (left) and LFQ (right). A. Scatter plot showing the log_2 fold change (FC) in comparison to parental wild type of mRNA and proteins encoded on all monosomes (blue), and all disomes (yellow). The marginal density histograms show the distribution of respective mRNAs and proteins. The expected median fold change of monosomic genes is marked by red dashed lines. The measured median fold changes of monosomic and disomic genes is marked by blue and yellow dashed lines, respectively. Note that the monosomy X was not evaluated by LFQ. B. Overlays of mRNA

(dashed line) and protein (solid line) density histograms. Values of respective medians are plotted in the graph. Left: TMT, right: LFQ. C. Two-dimensional annotation (2D) enrichment analysis based on the protein and mRNA changes in the monosomic cell line RM13 compared to RM 10;18. Each dot represents one category (GOBP, GOCC and chromosome location). Colors mark groups of related pathways as described in the inset. Axis-position represents scores of the pathways; negative values indicate downregulation, positive values indicate upregulation. Benjamini-Hochberg FDR Threshold < 0.02; left: TMT, right: LFQ.

More information on the TMT labeling needs to be provided. At minimum, one would want to know which TMT label was used for which sample. It is unclear if replicates of the cell lines were multiplexed or if the cell lines were multiplexed etc. Perhaps, much of this is in the PRIDE submission but it should at least be possible to understand the experimental design of the proteome analysis.

>> Indeed, this information can be acquired from the PRIDE database, but we now added it to Material and Methods.

Reviewers' Comments:

Reviewer #1:

Remarks to the Author:

The authors have addressed some of my concerns. However, there are still major concerns that are closely related to the validity of the conclusion of the paper.

First, bias may be introduced in the quantification of genes located on the monosomic chromosomes during normalization, because genes on these chromosomes have very different distribution compared to other genes. A simple median normalization may lead to over-estimation of protein abundance of genes on the monosomic chromosomes, leading to the reported buffering effect. I appreciate the addition of the LFQ data, but LFQ data normalization shares the same problem. The authors will need to generate targeted absolute quantification data (e.g., qPCR for RNA and SRM for proteins) for genes on the monosomic chromosomes to validate the reported buffering effect.

Second, the conflict between buffering at the protein level and ribosome deprivation needs to be further explored. Ribosome deprivation will only make buffering more difficult. Although protein level buffering can also be explained by reduced protein degradation, this is highly speculative without any supporting data.

Reviewer #2:

Remarks to the Author:

In this revised version, the authors renamed the manuscript and toned down the conclusion because of the difficulty to prove the link between RP gene haploinsufficiency and monosomy. However, the abstract is still the same as in the previous version. The authors need to adequately edit the abstract as well. In addition, the latter half of Discussion (p.11-p.12) also has to be extensively edited according to the new title.

Minor comment

p.7, line 8: Sentences are missing.

Reviewer #4:

Remarks to the Author:

The authors have adequately addressed my concerns.

It is a weak argument though to state that many publications use MS2 for TMT quantification. It is well established that TMT produces non-zero values even in MS3 spectra when the values should really be zero. No LC separation will change this. The question is how large the issue is.

Reviewer #5:

Remarks to the Author:

As a new reviewer who was asked to focus on the responses to comments from reviewer #3, it appears that the authors have addressed the major concerns and explained any misunderstandings that arose in the original review.

REVIEWER COMMENTS

Reviewer #1 (Remarks to the Author):

The authors have addressed some of my concerns. However, there are still major concerns that are closely related to the validity of the conclusion of the paper.

First, bias may be introduced in the quantification of genes located on the monosomic chromosomes during normalization, because genes on these chromosomes have very different distribution compared to other genes. A simple median normalization may lead to over-estimation of protein abundance of genes on the monosomic chromosomes, leading to the reported buffering effect. I appreciate the addition of the LFQ data, but LFQ data normalization shares the same problem. The authors will need to generate targeted absolute quantification data (e.g., qPCR for RNA and SRM for proteins) for genes on the monosomic chromosomes to validate the reported buffering effect.

>>To address this concern, we measured mRNA and protein levels of several candidates encoded on different monosomes by qPCR and western blotting (WB), respectively. In the analysis, we included examples of strongly compensated factors (e.g. BUB3, CDK1), as well as factors that showed almost no compensation (e.g. LAMP1, RPL21). All experiments were performed in several independent replicates different from the RNA and protein preps used for the TMT and LFQ analysis. As can be seen in figure R1, the mRNA levels analyzed by qPCR largely agree well with those determined by RNAseq, and the protein abundances estimated by WB corresponds with the measurements obtained by TMT mass spectrometry. By statistical analysis (Student's t test), there was no significant difference between TMT and WB, and between qPCR and RNAseq for individual factors. In all analyzed cases the trend of dosage compensation (or lack thereof) was identical regardless of the used method.

Additionally, we have applied several different types of normalization for both TMT and LFQ data, either including the monosomic chromosomes or separately for monosomes and disomes, always with the same conclusion, namely that a significant fraction of proteins encoded on the monosomes is expressed at an abundance higher than expected. Importantly, the normalization factor for each measurement the LFQ algorithm is optimized in order to achieve the least possible amount of differential regulation of most proteins. Since in both diploids, but also in monosomic cells most proteins are located on the disomes, it is difficult to imagine that such a normalization should introduce selectively a bias to the monosomically encoded proteins. While the distributions of mRNA and protein levels for monosomically encode proteins may differ of those of the disomically encoded, it is unclear to what degree this would bias the level of compensation we describe here. Given the good agreement observed for the tested cases between our omics data and data generated by qPCR and immunoblotting, the magnitude of a bias (if any) appears to be small.

Second, the conflict between buffering at the protein level and ribosome deprivation needs to be further explored. Ribosome deprivation will only make buffering more difficult. Although protein level buffering can also be explained by reduced protein degradation, this is highly speculative without any supporting data.

>> We do not see a conflict here. In our manuscript, we showed that monosomy results in a reduced rate of translation by puromycin incorporation assay, but it does not lead to a reduction in global total protein levels (as evidenced by mass spectrometry). This means that translation occurs at a slower rate, but it does not necessarily lead to lower protein abundance. Therefore, the protein levels could be also compensated by selective increase in translation, although at slower rates.

Moreover, the buffering affects only a small fraction of specific proteins (No. of genes compensated at protein level: 108 for chr. 10, 55 for chr. 13, 29 for chr. 18 and 86 for 19p). Additionally, many of the factors appear to be buffered on mRNA level (no. of genes compensated at mRNA level: 102 for chr.10, 12 for chr.13, 42 for chr.18 and 51 for chr.19p) (Fig. 3e). Several, not mutually exclusive mechanisms can be responsible for the buffering: increased transcription, increased stability of mRNA, differential translational regulation, or reduced protein degradation. At this point we indeed do not know the possible mechanism. Addressing this question is another project beyond the scope of this manuscript. This is also reflected in the discussion, where we just state the observation, but do not claim to know the reason.

Reviewer #2 (Remarks to the Author):

In this revised version, the authors renamed the manuscript and toned down the conclusion because of the difficulty to prove the link between RP gene haploinsufficiency and monosomy. However, the abstract is still the same as in the previous version. The authors need to adequately edit the abstract as well. In addition, the latter half of Discussion (p.11-p.12) also has to be extensively edited according to the new title.

>> We edited the abstract as recommended and appropriate in respect to the change of the focus. Regarding the Discussion, we also made some changes as recommended, although not extensive. In our opinion, the Discussion is well compatible with the new title. For example, the aspect of ribosome haploinsufficiency was mentioned only once in the connection to monosomy, and we clearly stated that this remains a hypothesis to be tested in the future.

Minor comment

p.7, line 8: Sentences are missing.

>> We thank the reviewer for careful reading. We have now added the sentences that were accidentally deleted.

Reviewer #4 (Remarks to the Author):

The authors have adequately addressed my concerns.

It is a weak argument though to state that many publications use MS2 for TMT quantification. It is well established that TMT produces non-zero values even in MS3 spectra when the values should really be zero. No LC separation will change this. The question is how large the issue is.

>> We are glad that we were able to address the concerns. We agree with the comment. We did our best with adding the LFQ measurements, with identical results, suggesting that the magnitude of a bias (if any) is rather small.

Reviewer #5 (Remarks to the Author): Expert in chromosomal instability and aneuploidies

As a new reviewer who was asked to focus on the responses to comments from reviewer #3, it appears that the authors have addressed the major concerns and explained any misunderstandings that arose in the original review.

>> We are glad that we were able to address the concerns and explain the misunderstandings.

Reviewers' Comments:

Reviewer #1:

Remarks to the Author:

The authors have adequately addressed my concerns.